# Gene expression dysregulation domains are not a specific feature of Down syndrome

Helena Ahlfors[1,6], Nneka Anyanwu[2,6], Edvinas Pakanavicius[2], Natalia Dinischiotu[2], Eva Lana-Elola[2], Sheona Watson-Scales[2], Justin Tosh[3], Frances Wiseman [3], James Briscoe [2], Karen Page[4], Elizabeth M.C. Fisher [3] & Victor L.J. Tybulewicz [2,5]

Down syndrome (DS), trisomy of human chromosome 21 (Hsa21), results in a broad range of phenotypes. A recent study reported that DS cells show genome-wide transcriptional changes in which up- or down-regulated genes are clustered in gene expression dysregulation domains (GEDDs). GEDDs were also reported in fibroblasts derived from a DS mouse model duplicated for some Hsa21-orthologous genes, indicating cross-species conservation of this phenomenon. Here we investigate GEDDs using the Dp1Tyb mouse model of DS, which is duplicated for the entire Hsa21-orthologous region of mouse chromosome 16. Our statistical analysis shows that GEDDs are present both in DS cells and in Dp1Tyb mouse fibroblasts and hippocampus. However, we find that GEDDs do not depend on the DS genotype but occur whenever gene expression changes. We conclude that GEDDs are not a specific feature of DS but instead result from the clustering of co-regulated genes, a function of mammalian genome organisation.

[1] NE Thames Regional Genetics Laboratory, GOSH NHS Foundation Trust, London WC1N 3JH, UK. [2] The Francis Crick Institute, London NW1 1AT, UK. [3] UCL Institute of Neurology, London WC1N 3BG, UK. [4] Department of Mathematics, University College London, London WC1E 6BT, UK. [5] Imperial College, London W12 0NN, UK. [6] These authors contributed equally: Helena Ahlfors, Nneka Anyanwu. Correspondence and requests for materials should be addressed to E.M.C.F. (email: elizabeth.fisher@ucl.ac.uk) or to V.L.J.T. (email: Victor.T@crick.ac.uk)

own syndrome (DS), also known as trisomy 21, is a leading cause of cognitive deficits, occurring in 1 in 700 births. DS results in a broad range of phenotypes, including cognitive impairment, congenital heart defects, craniofacial abnormalities and early-onset dementia[1]. The predominant view is that these phenotypes result from an increased dosage of one or more of the genes on Hsa21; currently, Hsa21 is estimated to contain 234 protein-coding genes[2]. The increased dosage of these genes is predicted to lead to increased transcript and protein levels (for the coding genes), which in turn would affect cellular and organismal physiology resulting in the observed pathologies. One DS pathological mechanism may be through the action of proteins such as transcription factors or chromatin modifiers that alter expression of other non-Hsa21 genes. A notable recent study proposed that trisomy 21 results in genome-wide transcriptional changes in which upregulated or downregulated genes are clustered in regions termed gene expression dysregulation domains (GEDDs), with genes whose expression changes in the same direction (up or down) being clustered[3]. The study postulated that GEDDs were the result of genome-wide chromatin changes in DS, potentially caused by overexpression of a chromatin modifier on Hsa21. The study also reported that GEDDs were present in a mouse model of DS named Ts65Dn that is trisomic for 132 protein-coding genes on mouse chromosome 16 (Mmu16) that are orthologous to Hsa21. However, the published study was limited in scope because GEDDs were identified using four replicate RNA samples derived from three independent fibroblast cultures isolated from a single pair of monozygotic twins discordant for trisomy 21, and the analysis of mouse Ts65Dn fibroblasts was carried out using just one replicate[3].

Evidently, such genome-wide changes in gene expression could make a significant contribution to DS phenotypes and thus merit further investigation. We examined the phenomenon of GEDDs using a recently created mouse model of DS termed Dp1Tyb, which has three copies of the entire 23 Mb region of Mmu16 that is orthologous to Hsa21 containing 148 protein-coding genes, including all the 132 genes duplicated in Ts65Dn[4]. Dp1Tyb mice show a number of DS-like phenotypes, including congenital heart defects[4], locomotor defects[5], learning and memory deficits and craniofacial abnormalities (Eva Lana-Elola, Sheona Watson-Scales, Elizabeth M.C. Fisher and Victor L.J. Tybulewicz, 2019 unpublished).

Here we develop a robust statistical method to evaluate whether changes in gene expression are more clustered than expected by chance, a necessary feature of GEDDs. Using this approach, we are indeed able to detect GEDDs in both human DS fibroblasts and in Dp1Tyb mouse fibroblasts and hippocampus. However, we show that the presence of GEDDs does not depend on genotype (for example, DS in humans or the Dp1Tyb duplication in mouse) but is seen whenever gene expression changes. Indeed, we detect GEDDs in gene expression data sets with no relation to DS or mouse models of DS. Furthermore, we show that the boundaries of GEDDs correlate with boundaries of topologically associating domains (TADs), units of higher-order chromatin structure that are enriched in co-regulated genes[6,7]. We conclude that GEDDs are not a specific feature of DS but instead result from the organisation of mammalian genomes whereby genes located close to each other are more likely to be co-regulated.

## Results

### Differential gene expression in Dp1Tyb mouse embryonic fibroblasts (MEFs).
The report of GEDDs in gene expression data from human DS fibroblasts and DS induced pluripotent stem cells (IPSCs) and from mouse Ts65Dn fibroblasts (a model of DS)[3]

presented an opportunity to use mouse genetics to map and identify the dosage-sensitive gene(s) causing this phenomenon. We decided to make use of the recently described Dp1Tyb mouse model of DS, which carries a duplication of a 23 Mb region of Mmu16 that is orthologous to Hsa21 and contains 148 protein-coding genes[4]. This mouse strain has been backcrossed for well over 10 generations such that the genetic background of all animals is C57BL/6J with <0.1% genetic variability between mice. This duplicated region in Dp1Tyb includes all 132 Mmu16 genes duplicated in Ts65Dn but does not have increased dosage of any of the 43 coding genes non-orthologous to Hsa21 that are duplicated in Ts65Dn[8]. Thus, if GEDDs are caused by increased dosage of Hsa21 genes or their orthologues on Mmu16 in the mouse, they should also be seen in cells from Dp1Tyb mice. The Dp1Tyb strain offers a further advantage in that we have also generated a series of strains with shorter nested duplications on Mmu16 (Dp2Tyb to Dp9Tyb), allowing mapping of the causative gene(s) and their eventual identification[4].

Since GEDDs had been reported in gene expression data from human DS fibroblasts and Ts65Dn MEFs, we chose to analyse Dp1Tyb MEFs. We grew cultures of MEFs from four wild-type (WT) and five Dp1Tyb littermate embryos, isolated RNA from them and carried out RNA sequencing (RNAseq) in order to quantitate gene expression. Differential gene expression analysis showed 66 significantly differentially expressed genes (adjusted $p$ value $[p_{adj}] < 0.05$) with 39 of these located within the duplicated region in Dp1Tyb, all of which were upregulated (Fig. 1, Supplementary Data 1). The mean fold change of these duplicated genes in Dp1Tyb vs WT MEFs was 1.47, close to the expected value of 1.5.

**GEDDs in Dp1Tyb MEFs**. The existence of GEDDs was previously proposed by analysing the fold change in gene expression of DS fibroblasts or IPSCs vs euploid cells and looking for clustering of upregulated or downregulated genes across the genome[3]. In this study, the visualisation of these domains of clustered gene expression changes was simplified using a Loess smoothed curve. However, the study did not determine whether the clustering of gene expression changes seen in human DS cells was statistically significant. We employed a similar approach to the gene expression data from Dp1Tyb MEFs to look for GEDDs. As expected, we could see a clear increase of about 1.5-fold in gene expression in Dp1Tyb MEFs across the duplicated region of Mmu16 (Fig. 2). Across the rest of the genome, however, the fold changes were small (Fig. 2, Supplementary Fig. 1). Notably, the changes in gene expression were much smaller than those previously reported for the comparison between DS and euploid human fibroblasts that had led to the concept of GEDDs (Supplementary Fig. 2a)[3]. These larger fold changes in the human data are most likely the consequence of much greater variation in gene expression seen in the RNAseq data (Supplementary Fig. 2b). Nonetheless, in the analysis of the RNAseq data from Dp1Tyb MEFs, the Loess smoothed curves showed regions of apparent clustered upregulation or downregulation. However, the extent of upregulation or downregulation was small, and from this analysis, it was not possible to determine whether such clusters were statistically significant.

To address this issue, we devised two statistical tests to determine whether upregulated or downregulated genes were clustered more than would be expected by chance, a necessary feature of GEDDs. First, for each chromosome we counted the numbers of flips defined as a change in the direction of the fold change from one gene to the adjacent gene on the chromosome (Fig. 3a). This number of flips was compared to the distribution of numbers of flips determined from 100,000 permutations of the

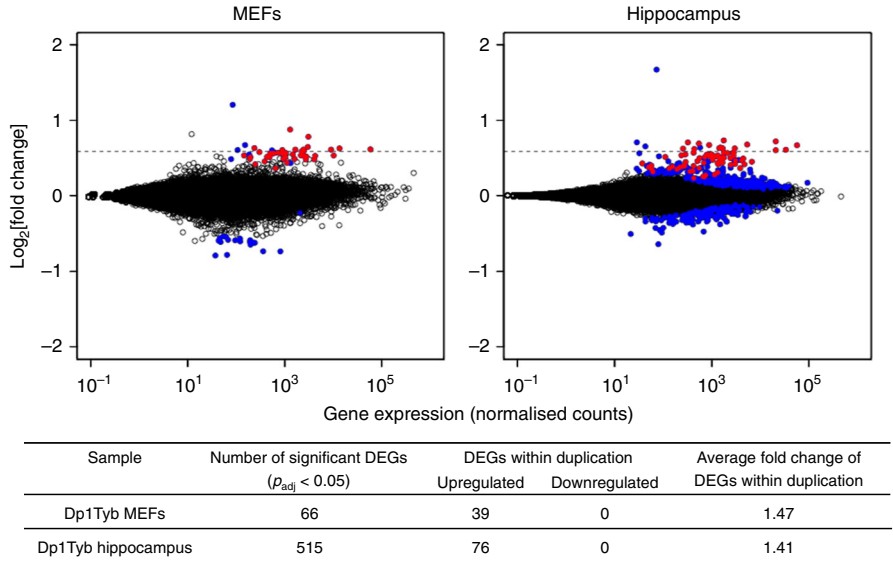

**Fig. 1** Differential gene expression in Dp1Tyb mice. Plots of log$_2$[fold change] in the expression of each gene in Dp1Tyb mouse embryonic fibroblasts (MEFs, left) or hippocampus (right) compared to wild-type (WT) littermate controls, against mean expression of each gene. Genes are indicated by black circles, except for differentially expressed genes (DEGs, $p_{adj} < 0.05$) indicated in red if they are within the duplicated region and in blue if outside it. Dashed lines indicate a fold change of 1.5. Table at bottom shows the number of DEGs, the number of DEGs within the duplicated region in Dp1Tyb (all of which were upregulated) and the average fold change of the latter

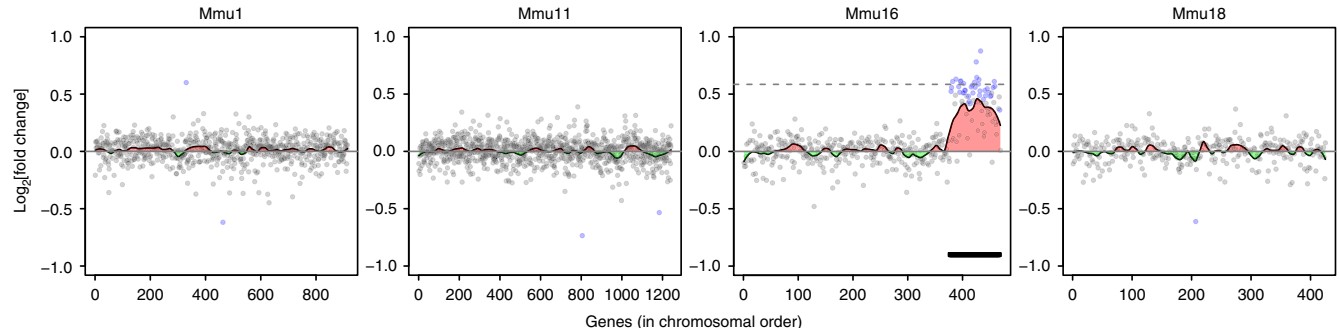

**Fig. 2** Change in expression in Dp1Tyb mouse embryonic fibroblasts (MEFs) as a function of chromosomal position. Plots show fold change of gene expression in Dp1Tyb MEFs vs wild-type control cells. Expressed genes are plotted in chromosomal order on the four example chromosomes. Genes that are significantly differentially expressed are indicated with blue dots, other genes are in grey. A Loess smoothing curve is superimposed with regions that are upregulated or downregulated indicated in red or green. Dashed line on Mmu16 indicates a fold change of 1.5. Note the upregulation of genes in the duplicated region on Mmu16 (thick black line)

same chromosome with randomised orders of the genes. Clustering of upregulated or downregulated genes would result in a significant decrease in the number of flips compared to the random (bootstrapped) distribution. Second, we used a spatial correlation measure[9]. To this end, we calculated an energy function $E$ for each chromosome by multiplying the fold change of each gene with each of its two neighbours and then adding these products together for all genes on the chromosome (Fig. 3a). This measure takes into account the magnitude as well as the direction of the change of each gene and would be increased if there were more clustering of upregulated or downregulated genes. Again, this measure was compared to the distribution of $E$ of 100,000 bootstrapped versions of the same chromosome. The presence of GEDDs would result in a larger value of $E$ than expected by chance.

We undertook both tests (numbers of flips, energy function) on the human DS fibroblast gene expression data in Letourneau et al., which had reported the existence of GEDDs[3]. On most

chromosomes, we were indeed able to detect significantly decreased numbers of flips and increased energy compared to the numbers expected by chance (>2 standard deviations (SDs) from mean of randomised distribution), validating these two approaches, and confirming the presence of GEDDs statistically (Fig. 3b, Supplementary Fig. 3, Supplementary Fig. 4, Table 1). Next, we examined the RNAseq data from Dp1Tyb and WT MEFs. Here again we found significantly decreased numbers of flips and increased energy on most chromosomes, implying that GEDDs were also present in Dp1Tyb MEFs (Fig. 3c, Supplementary Fig. 5, Supplementary Fig. 6, Table 1).

**GEDDs in Dp1Tyb hippocampus.** The analysis by Letourneau et al. showed a correlation in the distribution of GEDDs between those seen in human DS and mouse Ts65Dn fibroblasts[3]. Thus we examined whether the same was true for gene expression changes detected in Dp1Tyb fibroblasts. However, excluding genes on

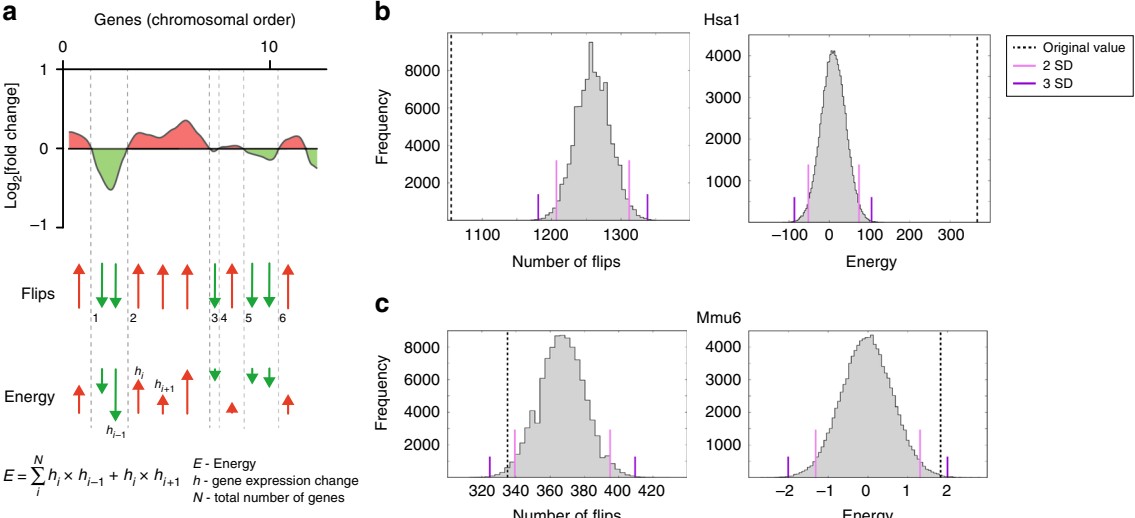

**Fig. 3** Statistical tests for gene expression dysregulation domains. **a** An example plot of fold change of gene expression against chromosomal position showing regions in which expressed genes are upregulated or downregulated (red and green, respectively). Below this are shown the directions of the fold changes (red and green arrows) with a dashed black line indicating a flip, i.e. a change in direction of the fold change. In the example shown, there are six flips. The energy function (E) is calculated by taking into account both the direction and magnitude of the fold change and is defined as the sum over all genes on the chromosome of the products of the fold change (h) of a given gene (i) with the fold change of the previous gene (i–1) and the same product with the next gene (i+1). **b** Distribution of flips and energy in 100,000 versions of, for example, Hsa1 each with a different randomised order of genes from the human Down syndrome fibroblast expression data[3]. Pink and purple lines indicate 2 and 3 standard deviations (SD) away from the mean, respectively; dashed line shows the flips and energy for the actual non-randomised chromosome. **c** Distribution of flips and energy as described in **b**, for example chromosome Mmu6 in gene expression data from Dp1Tyb and wild-type mouse embryonic fibroblasts

**Table 1 Human DS fibroblasts[3] and mouse Dp1Tyb MEFs and hippocampus show GEDDs**

| Sample | Number of significant chromosomes | |
| --- | --- | --- |
| | **Flips** | **Energy** |
| Human DS fibroblasts | 20 | 22 |
| Dp1Tyb MEFs | 7 | 16 |
| Dp1Tyb hippocampus | 8 | 17 |

Table shows the numbers of chromosomes that had significantly (>2 SD) reduced numbers of flips or increased energy in comparisons of changes in gene expression in human DS vs euploid fibroblasts or in Dp1Tyb vs WT MEFs or hippocampus
DS, Down syndrome; GEDD, gene expression dysregulation domain; MEF, mouse embryonic fibroblast

Hsa21 and their orthologues on Mmu16, we were able to detect only a very small correlation in the expression changes between orthologous genes in human DS and mouse Dp1Tyb fibroblasts (Fig. 4a). In view of this and the very small changes in gene expression we saw between Dp1Tyb and WT MEFs (only 27 differentially expressed genes outside the duplicated region), we wondered whether another cell type might show larger differences in gene expression, making the correlation between human and mouse gene expression changes easier to detect.

We decided to evaluate the gene expression changes in hippocampus from five WT and five Dp1Tyb mice, since this region of the brain plays an important role in learning and memory and its function is altered in several mouse models of DS[10–17], including Dp1Tyb (Elizabeth M.C. Fisher and Victor L.J. Tybulewicz, 2019 unpublished). Compared to the MEF expression data, we saw more changes in gene expression in the hippocampus. There were 515 significantly differentially expressed genes with 76 of these located within the duplicated region in Dp1Tyb, all of which were upregulated, and a further

439 differentially expressed genes outside the duplication (Fig. 1, Supplementary Data 2). The mean fold change of the duplicated genes in Dp1Tyb vs WT hippocampus was 1.41, once again close to the expected value of 1.5. Plots of fold change of genes along chromosomes also showed clear upregulation of genes across the duplicated region with only small changes across the rest of the genome (Fig. 5, Supplementary Fig. 7). Once again, the fold changes in gene expression were much smaller than those previously reported for the comparison between DS and euploid human fibroblasts[3], most likely due to much higher variation in gene expression in the human data (Supplementary Fig. 2a, b). Moreover, analysis of clustering of gene expression changes showed many chromosomes with significantly decreased numbers of flips and increased energy confirming that GEDDs were also detectable in the Dp1Tyb hippocampus (Supplementary Fig. 8, Supplementary Fig. 9, Table 1).

The study reporting GEDDs in human DS fibroblasts showed that these domains were also seen in DS IPSCs with substantial correlation in the location and magnitude of GEDDs between these two different cell types[3]. Thus we examined the correlation in gene expression changes between the Dp1Tyb mouse hippocampus and Dp1Tyb MEFs. Excluding the duplicated genes on Mmu16, we were able to detect only very weak correlation (Fig. 4a). This lack of correlation was also seen in an overlay of the Loess curves of the RNAseq data from Dp1Tyb MEFs and hippocampus (Supplementary Fig. 10). Furthermore, there was also very low correlation in the gene expression changes between Dp1Tyb hippocampus and human DS fibroblasts (Fig. 4a). Taken together, these data show that GEDDs can be detected in Dp1Tyb MEFs and hippocampus using sensitive statistical tests, but as shown by the very poor correlation in expression changes, the location of these is not conserved between human DS and mouse models or indeed between different tissues in the Dp1Tyb mouse strain.

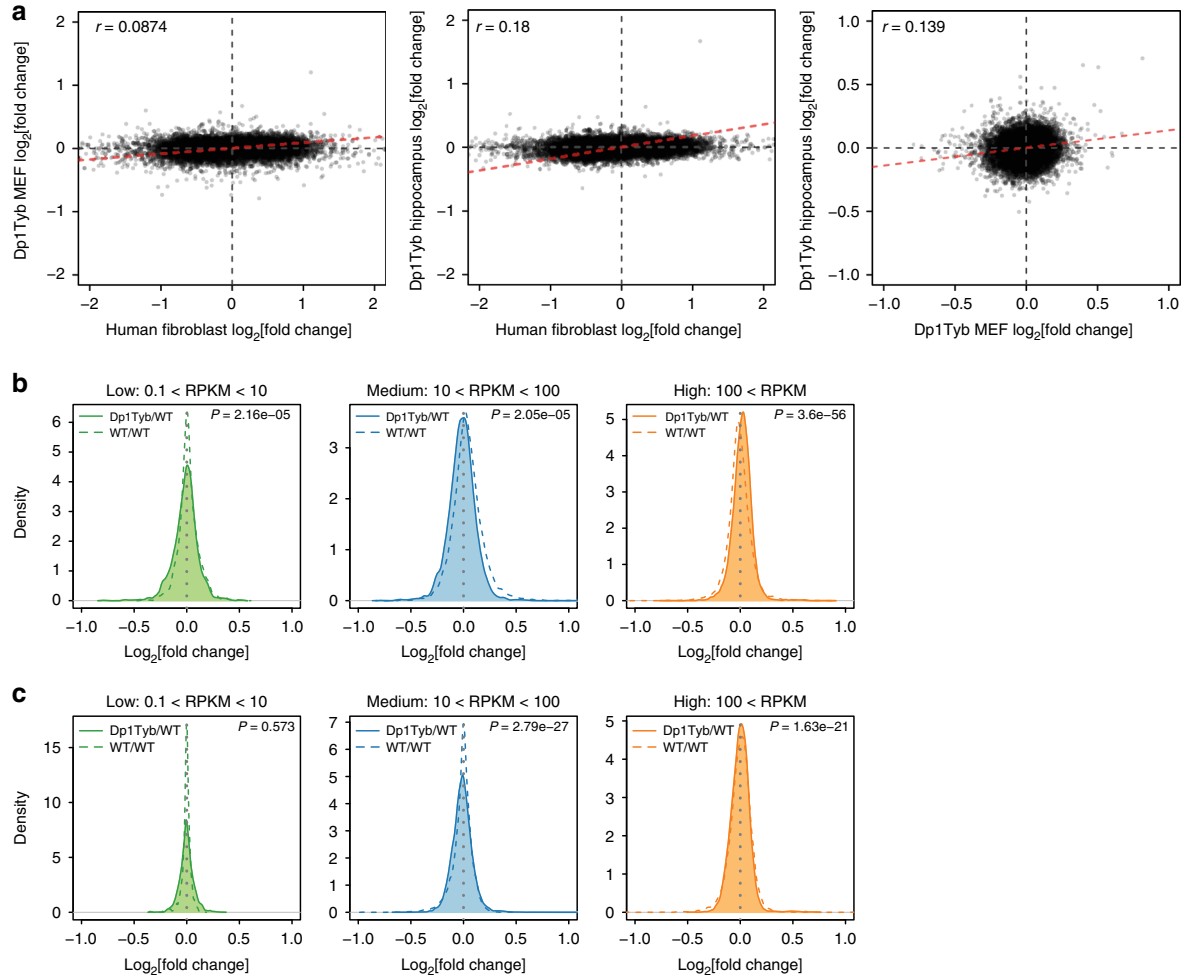

**Fig. 4** Very little correlation in gene expression changes between human Down syndrome (DS) and mouse Dp1Tyb fibroblasts or hippocampus and no preferential upregulation of lower expressed genes. **a** Correlation plots showing the fold change in gene expression comparing human DS vs euploid fibroblasts with Dp1Tyb vs wild-type (WT) mouse embryonic fibroblasts (MEFs) or hippocampus or comparison of the Dp1Tyb MEFs with hippocampus. Each dot is a gene (in human vs mouse comparisons orthologous genes were used). Dashed red line indicates best fit regression line. Correlation coefficient *r* is indicated for each comparison. Hsa21 genes and their mouse orthologues were excluded from the analysis. **b**, **c** Density plots of fold change in gene expression between Dp1Tyb vs WT MEFs (**b**) or hippocampus (**c**) (solid line) divided according to the level of gene expression: low (0.1 < RPKM < 10), medium (10 < RPKM < 100), high (100 < RPKM). Dashed line shows similar plots of a comparison of gene expression between two WT samples and two other WT samples. Statistical significance was calculated using Mann–Whitney test

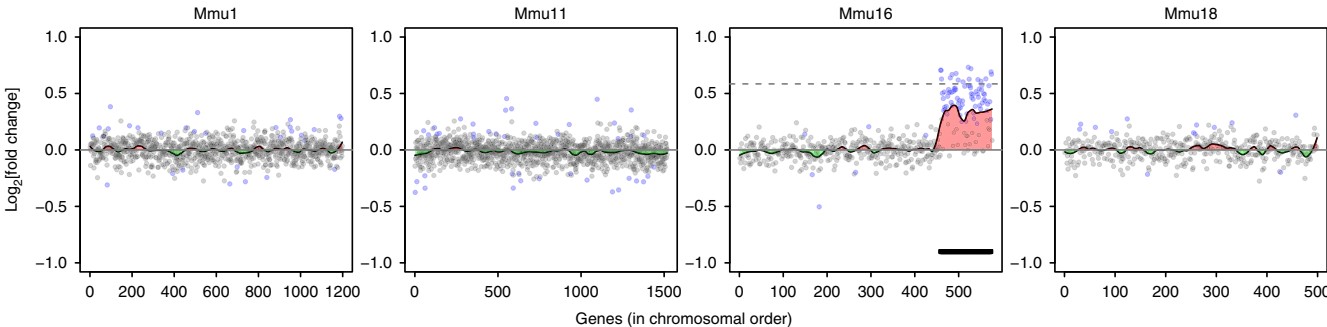

**Fig. 5** Change in expression in Dp1Tyb hippocampus as a function of chromosomal position. Plots show fold change of gene expression in Dp1Tyb vs wild-type hippocampus. Expressed genes are plotted in chromosomal order on the four example chromosomes. Genes that are significantly differentially expressed are indicated with blue dots, other genes are in grey. A Loess smoothing curve is superimposed with regions that are upregulated or downregulated indicated in red or green, respectively. Dashed line on Mmu16 plot indicates a fold change of 1.5. Note the upregulation of genes in the duplicated region on Mmu16 (thick black line)

**GEDDs are not caused by a decreased dynamic range of gene expression**. Letourneau et al. reported that, in the comparison of DS and euploid fibroblasts, DS cells had elevated expression of genes expressed at a low level and decreased expression of more highly expressed genes[3]. Since genes tend to be clustered according to level of expression, the authors suggest that GEDDs may arise because of a smaller dynamic range of gene expression in DS cells compared to euploid cells, leading to clustered increases in gene expression of lowly expressed genes and clustered decreases of highly expressed genes. In view of this, we examined the gene expression changes in Dp1Tyb MEFs and hippocampus compared to WT controls as a function of gene expression level. Following the approach used by Letourneau et al.[3], we divided genes into low, medium and high levels of expression and evaluated the distribution of fold changes in gene expression between Dp1Tyb and WT MEFs or hippocampus in comparison to the control of comparing WT to WT expression. We saw no evidence for increased or decreased fold changes in Dp1Tyb cells in the lowly or highly expressed genes, respectively (Fig. 4b, c). Thus there is no change in the dynamic range of gene expression in Dp1Tyb MEFs or hippocampus, and this cannot explain the GEDDs in these cell types.

**GEDDs are not due to increased mRNA levels in Dp1Tyb cells**. A recent study by Mowery et al. demonstrated that interleukin-7-cultured pro-B cells from Ts1Rhr mice had increased levels of mRNA compared to WT controls[18]. This mouse strain has a duplication of a 33-gene Hsa21-orthologous region on Mmu16, which is entirely included within the duplication in Dp1Tyb mice[4,19]. The study demonstrated that this increase in mRNA is caused by an additional copy of the *Hmgn1* gene, which codes for HMGN1, a nucleosome-binding protein that modulates chromatin compaction. Furthermore, the increase in mRNA level was not even, with a larger increase in lower expressed genes and a smaller increase in the more highly expressed genes. The authors argue that, in a standard RNAseq analysis that assumes no change in overall RNA levels and is normalised to median read counts (relative normalisation), this uneven increase in RNA levels would lead to an apparent increase in the expression of lower expressed genes and decrease in the expression of higher expressed genes. This in turn would result in the appearance of GEDDs because of the tendency for genes to be clustered according to the level of expression. Since the *Hmgn1* gene is duplicated in Dp1Tyb mice and significantly increased in the expression in both Dp1Tyb MEFs and hippocampus (Supplementary Data 1, 2), it is possible that the GEDDs we observed in tissues from these mice might also be a consequence of increased mRNA levels and the use of relative normalisation. To address this possibility, we carried out another RNAseq experiment on Dp1Tyb and WT MEFs, but this time added non-mammalian synthetic ERCC (External RNA Controls Consortium) RNA controls at a fixed amount per cell to each sample, similar to the strategy employed by Mowery et al.[18]. To determine whether the overall amount of mRNA was increased in Dp1Tyb MEFs, we analysed the data by normalising to the spiked-in ERCC controls (absolute normalisation) and compared this to relative normalisation of the same data. We found that the mean fold change of gene expression between Dp1Tyb and WT MEFs was ~1.8% higher using absolute normalisation compared to relative normalisation, indicating a small increase in overall mRNA level in Dp1Tyb MEFs (Supplementary Fig. 11a). This is much less than the ~10% increase seen by Mowery et al. in Ts1Rhr pro-B cells[18] and may be partly accounted for by the increase in the transcriptome in Dp1Tyb cells compared to WT cells (~0.7%). Importantly, we could see no skewed increase in expression in

favour of lower expressed genes (Supplementary Fig. 11b), thus an increase in mRNA level cannot explain the GEDDs detected in these cells.

**GEDDs are not caused by DS genotype**. In view of the very low correlation between the human and mouse gene expression changes and the very small magnitude of these changes in both Dp1Tyb MEFs and hippocampus, we wondered whether the expression changes contributing to the detection of GEDDs were due to experimental variation and not to the human or mouse DS genotype. To address this, we repeated the analysis of the expression data from both the human DS fibroblasts[3] and the mouse Dp1Tyb MEFs and hippocampus but mixed samples so as to eliminate the effect of genotype. The human DS fibroblast data consisted of four DS samples and four euploid samples. Thus we compared two DS and two euploid samples against two other DS and two other euploid samples, thereby eliminating the effect of genotype in the comparison. With 4 DS and 4 euploid samples, there are 18 possible combinations in which such no genotype difference comparisons could be carried out, and we calculated fold changes in gene expression for all of these. Similar switching was carried out with the Dp1Tyb MEF and hippocampus data.

Plotting the fold changes superimposed on Loess smoothing curves from the genotype-switched analysis of the human DS fibroblasts suggested the presence of GEDDs, for example on Hsa21 (Fig. 6a, b). Analysis of GEDDs using the flips and energy measures showed that, for many chromosomes in most of the switched no-genotype-difference analyses, there were significantly reduced numbers of flips and increased energy, indicating GEDDs (Fig. 6c, d). Similarly, eliminating the genotype contribution in the mouse Dp1Tyb MEF and hippocampus data by switching genotypes still showed that in most of the switched combinations most chromosomes had detectable GEDDs as measured by flips or energy (Fig. 7a–h, Supplementary Fig. 12 and 13). Thus we conclude that clustering of upregulated or downregulated genes (i.e. GEDDs) is detectable even when there is no difference in genotype between the samples being compared and, in this case, is most likely due to small variations in gene expression from one fibroblast culture to another or from one mouse to another. Furthermore, this implies that GEDDs may not be a specific feature of DS.

**GEDDs caused by genetic changes unrelated to DS**. If GEDDs are not specifically caused by the DS genotype but are detected when expression changes are due to other causes, it may be that they are a general phenomenon occurring whenever the expression of genes changes. To address this, we turned to a completely unrelated RNAseq data set that one of us (H.A.) had previously worked on, consisting of gene expression in follicular and marginal zone B cells taken from mice with a null mutation of the *Zfp36l1* gene or from WT control mice[20]. We carried out three differential gene expression analyses, comparing WT and ZFP36L1-deficient follicular B cells, WT and ZFP36L1-deficient marginal zone B cells and WT follicular with WT marginal zone B cells. All three of these comparisons identified substantial numbers of differentially expressed genes[20]. Plots of fold changes in gene expression with Loess smoothing curves indicated the possible locations of clustered gene expression changes (Supplementary Fig. 14, and data not shown). Furthermore, analysis of clustering of gene expression changes once again showed many chromosomes with significantly decreased numbers of flips and increased energy confirming that GEDDs were also detectable in each of these three comparisons (Table 2). Thus we conclude that GEDDs are caused by perturbations in gene expression that are

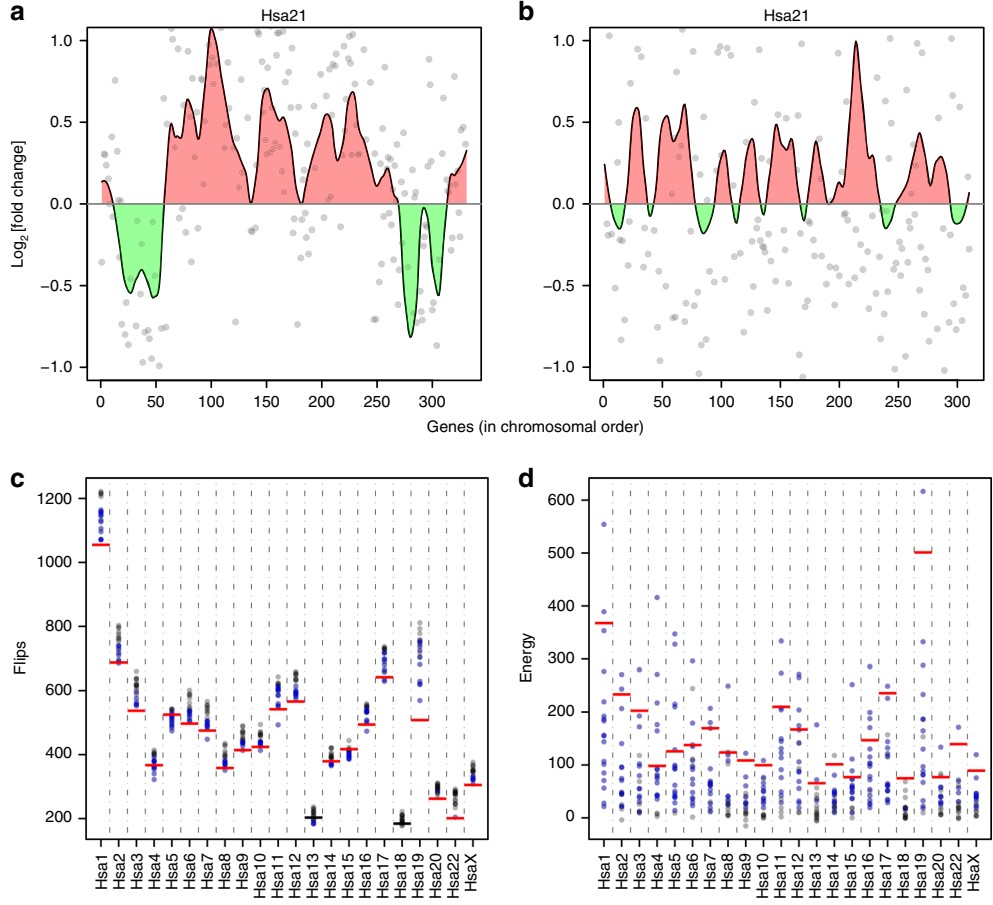

**Fig. 6** Gene expression dysregulation domains do not depend on the Down syndrome (DS) genotype. **a** Plot showing fold change of gene expression on Hsa21 in DS vs euploid human fibroblasts[3]. Expressed genes (grey dots) are plotted in chromosomal order. A Loess smoothing curve is superimposed with regions that are upregulated or downregulated indicated in red or green, respectively. **b** Same as **a** but the fold changes are computed between two DS and two euploid samples vs between two different DS and two different euploid samples. Note apparent regions of upregulation and downregulation in this no-genotype-difference comparison. **c**, **d** Plots of flips (**c**) and energy (**d**) for comparisons of different combinations of DS and euploid human fibroblast samples arranged such that there is no genotype difference in the comparison. Each dot is a comparison of different combinations of samples with blue indicating combinations that are significantly different (>2 SD) from the mean of bootstrapped chromosomes and black for not significant. Horizontal lines indicate the values obtained for the correct comparison of four DS vs four euploid samples (red, significant; black, not significant)

unrelated to DS and hence are not a specific feature of the syndrome.

**GEDD boundaries align with TAD boundaries**. Finally, we considered the relationship of the GEDDs that we had detected in Dp1Tyb MEFs and hippocampus with previously described genomic domains that relate to gene expression. TADs are regions of the genome showing increased intra-domain interactions compared to inter-domain interactions[6,7]. TADs appear to be loops of chromatin held together at their ends, one consequence of which is that genes within TADs are more likely to be co-regulated[21–23]. The boundaries of TADs are often marked by binding of the CTCF protein, which may be involved in the formation of TADs[6]. These boundaries also often correspond to the boundaries of replication domains, regions of either early or late replication[24]. Finally, some regions of the genome have been found to associate with the nuclear lamina. These lamina-associated domains (LADs) are typically regions of gene repression and correspond partially to TADs, though generally LADs are larger than TADs[25]. We compared the location of these elements relative to GEDDs and found that TAD boundaries and CTCF binding sites were enriched at GEDD boundaries from both Dp1Tyb MEFs and hippocampus (Fig. 8a, c).

In contrast, there was no obvious enrichment of replication domain or LAD boundaries. Inverting the comparison, we found that GEDD boundaries and CTCF-binding sites were enriched at TAD boundaries, as were replication domain and LAD boundaries, although to a lower extent (Fig. 8b, d). Thus we conclude that the co-regulation of gene expression in GEDDs is most likely a consequence of their partial correspondence to TADs, genomic structures that tend to contain co-regulated genes.

## Discussion

The recent report that DS cells show clustered upregulation or downregulation of gene transcription compared to euploid cells, a phenomenon termed GEDDs, gave rise to the hypothesis that at least some DS phenotypes may be caused by chromatin changes leading to genome-wide dysregulation of gene expression[3]. This is an important hypothesis to address in the context of trying to understand the pathological mechanisms that underpin DS phenotypes, since it suggests that some phenotypes may be the result of widespread changes of gene expression across the genome, rather than direct effects of increased expression of one or a few Hsa21 genes. The observation that GEDDs were also reported in the Ts65Dn mouse model of DS presented us with an opportunity to use the Dp1Tyb mouse strain and, potentially, the associated

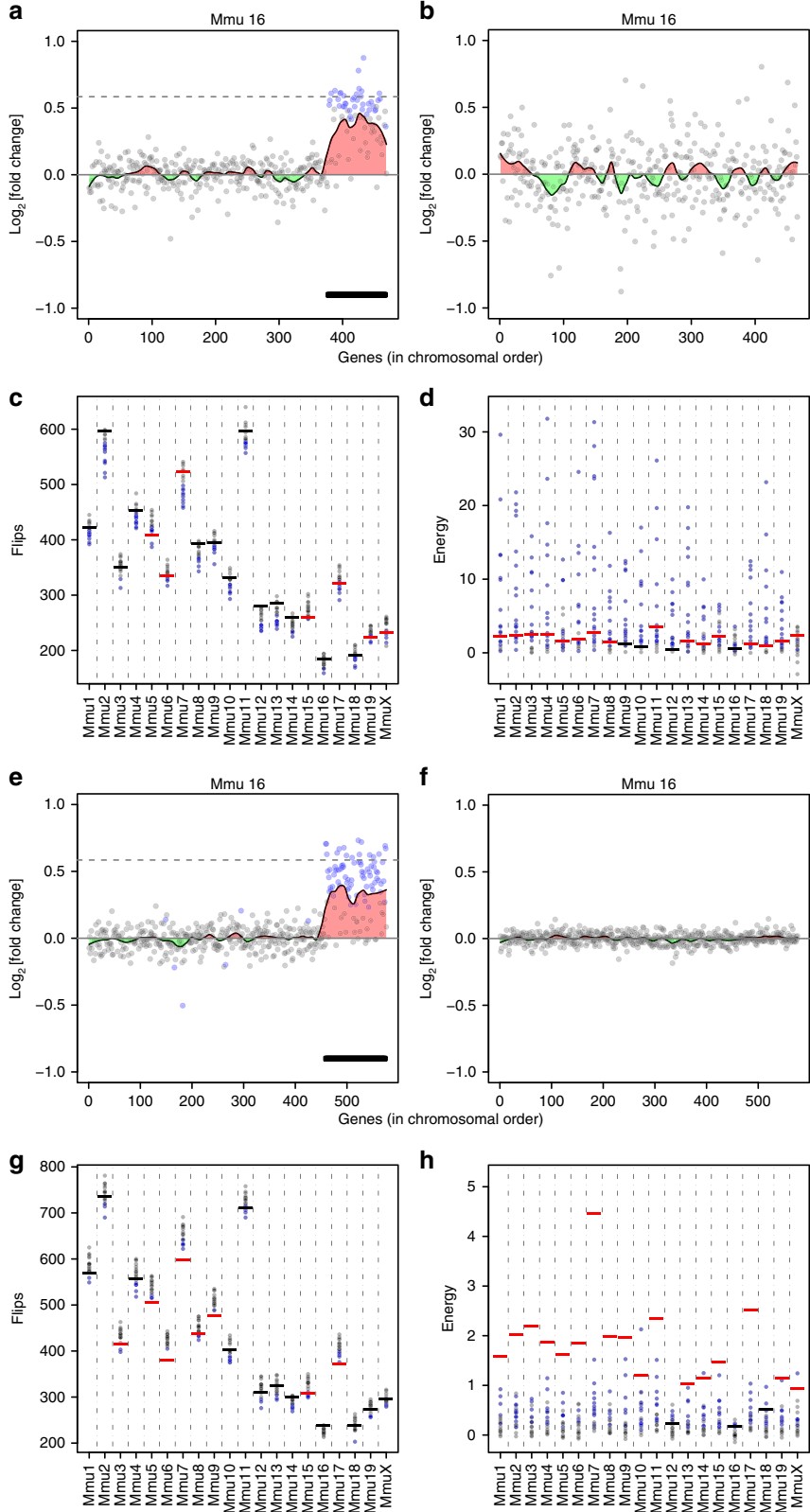

mapping panel of mouse duplication strains (Dp2Tyb to Dp9Tyb)[4] to locate and identify the dosage-sensitive gene(s) that give rise to GEDDs, and hence, ultimately, to test the importance of GEDDs in causing DS phenotypes.

We developed two statistical tests to confidently detect clustering of gene expression changes occurring in the same direction and using these were able to show GEDDs in the gene expression data from human DS fibroblasts reported by Letourneau et al.[3]. We were also able to detect GEDDs in our own gene expression data from Dp1Tyb MEFs and hippocampus. However, the location of the expression changes was not conserved between the human and mouse data. We noted that the magnitude of the gene

**Fig. 7** Gene expression dysregulation domains do not depend on the genotype of mouse models of Down syndrome (DS). **a** Fold change in gene expression on Mmu16 from a comparison of Dp1Tyb and wild-type (WT) mouse embryonic fibroblasts (MEFs). Expressed genes (grey dots) are plotted in chromosomal order. A Loess smoothing curve is superimposed with regions that are upregulated or downregulated indicated in red or green, respectively. Dashed line indicates a fold change of 1.5 and the thick black line indicates the duplicated region in Dp1Tyb mice. This is the same as the plot of Mmu16 in Fig. 2. **b** Same as **a**, but fold changes of gene expression on Mmu16 are from a no-genotype-difference comparison of two WT and two Dp1Tyb MEFs vs two other WT and two other Dp1Tyb MEFs. **c**, **d** Plots of flips (**c**) and energy (**d**) for comparisons of all possible no-genotype-difference combinations of WT and Dp1Tyb MEF samples. Each dot is a comparison of different combinations of samples, with blue indicating combinations that are significantly different (>2 SD) from the mean of bootstrapped chromosomes and black for not significant. Horizontal lines indicate the values obtained for the correct comparison of four DS vs four euploid samples (red, significant; black, not significant). **e** Fold change in gene expression on Mmu16 from a comparison of Dp1Tyb and WT hippocampus; identical to figure of Mmu16 shown in Fig. 5. **f** Same as **e**, but fold changes of gene expression on Mmu16 are from a no-genotype-difference comparison of two WT and two Dp1Tyb hippocampus vs two other WT and two other Dp1Tyb hippocampus. **g**, **h** Analysis of the distribution of gene expression changes in no-genotype comparisons for Dp1Tyb and WT mouse hippocampus data. Plots as in **c** and **d**

| Table 2 GEDDs observed in B cells null for ZFP36L1 | | |
|---|---|---|
| **Sample** | **Number of significant chromosomes** | |
| | **Flips** | **Energy** |
| ZFP36L1-deficient vs WT follicular B cells | 8 | 13 |
| ZFP36L1-deficient vs WT marginal zone B cells | 5 | 8 |
| Follicular vs marginal zone WT B cells | 4 | 10 |

Table shows the numbers of chromosomes that had significantly (>2 SD) reduced numbers of flips or increased energy in comparisons of changes in gene expression in ZFP36L1-deficient vs WT follicular or marginal zone B cells or in follicular vs marginal zone WT B cells GEDD, gene expression dysregulation domain; WT, wild type

expression changes was larger in the human DS fibroblast data compared to the Dp1Tyb mouse fibroblast and hippocampus data. This difference may arise from the greater number of biological replicates used in our mouse studies compared to the original human DS fibroblast data[3]. This is supported by the larger variation in gene expression, i.e. more noise, in the human fibroblast data compared to our Dp1Tyb mouse fibroblast and hippocampus data.

Letourneau et al. had previously suggested that GEDDs may be caused by a smaller dynamic range of gene expression in DS cells with increased expression of lower expressed genes and decreased expression of higher expressed genes[3]. We saw no such effect in Dp1Tyb MEFs or hippocampus. More recently, Mowery et al. showed that an additional copy of the *Hmgn1* gene results in an increase in total RNA in mouse pro-B cells, with a larger increase in lower than in higher expressed genes[18]. They suggested that, because of this, the appearance of GEDDs in DS cells was an illusion caused by the use of relative normalisation of RNAseq data to the median gene expression in a sample. To test this idea, we carried out RNAseq on Dp1Tyb and WT MEFs with control spike-in RNAs, which allows an absolute normalisation of the data to the spike-ins, thereby determining absolute RNA levels. We found a small increase in RNA levels in Dp1Tyb MEFs, but this increase was not skewed towards the lower expressed genes and thus cannot explain the GEDDs we observe. This result agrees with our observation, discussed above, that using a relative normalisation there was no increased expression of lower expressed genes and decreased expression of higher expressed genes. We note that, despite an additional copy of the *Hmgn1* gene in Dp1Tyb MEFs and ~1.5-fold increased expression of the gene, the increase in RNA level was much smaller than that seen in Ts1Rhr pro-B cells, suggesting that the effects of *Hmgn1* overexpression may be cell-type specific.

Importantly, using two different approaches, we were able to show that the presence of GEDDs was not dependent on the DS genotype. First, we mixed samples and carried out differential gene expression analysis between sets of samples that no longer differed by DS genotype; this analysis still detected GEDDs. Second, we were able to detect GEDDs in an unrelated RNAseq data set comparing gene expression in B cells from WT and ZFP36L1-deficient mice. Thus we conclude that GEDDs are not a specific feature of DS but rather this spatial coordination of expression in the genome is seen whenever gene expression changes, even if this is caused by small stochastic variations in gene expression.

A recent publication by Do et al. questioned the validity of GEDDs in DS, based on an inability to reproduce the gene expression changes in human IPSCs and Ts65Dn MEFs reported by Letourneau et al.[26]. This report points out that gene expression data on the same DS IPSCs had been previously published by the same group[27], but analysis of this latter data showed very little correlation in gene expression changes with the data in Letourneau et al.[26]. Furthermore, Do et al. also carried out RNAseq on Ts65Dn and control MEFs and showed that it did not correlate with the gene expression changes reported in Ts65Dn MEFs by Letourneau et al.[26]. Taken together, Do et al. suggest that GEDDs are not a reproducible feature of human DS IPSCs or of fibroblasts from the Ts65Dn mouse model of DS. These differences in gene expression changes may be a result of experimental variation; we note that the Letourneau et al.'s study used only three biological replicates for the RNAseq data from DS fibroblasts and only one replicate from DS IPSCs and Ts65Dn mouse fibroblasts[3].

Our data show that, irrespective of whether the reported gene expression changes are caused by the DS genotype or by experimental variation, whenever gene expression changes between two different conditions, the upregulated or downregulated expression changes are clustered to a greater extent than would be expected by chance. This clustering of gene expression changes is likely to be caused by the non-random arrangement of genes in the mammalian genome. It has been recognised for some time that co-expressed genes are more likely to be located near each other in the genome[28]. Co-expressed mammalian genes are clustered at different scales, both at the level of neighbouring genes and in domains spanning many megabases[29,30]. In particular, TADs are genomic domains that are formed by chromatin loops and are often bounded by CTCF-binding sites[6,7]. As a consequence of increased intra-domain interactions in TADs, genes within them are more likely to be co-regulated[21–23]. We found that the boundaries of TADs and CTCF-binding sites were enriched at the boundaries of GEDDs, suggesting that GEDDs at least partially correspond to TADs. This observation explains why we detect a statistically significant increase in clustering of gene expression changes.

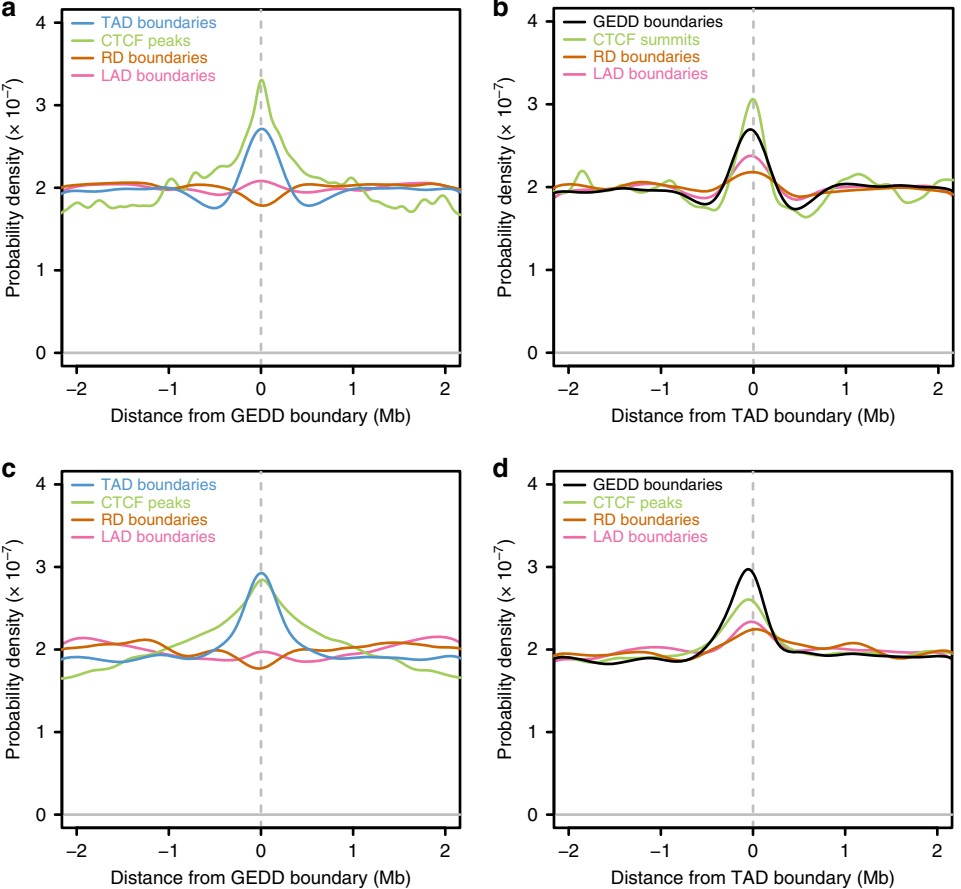

**Fig. 8** Boundaries of gene expression dysregulation domains (GEDDs) correlate with boundaries of topologically associating domains (TADs) and CTCF-binding sites. **a** Probability density plot showing the distribution of boundaries of TADs, lamina-associated domains (LADs) and replication domains (RDs) and CTCF-binding sites, all from mouse embryonic fibroblasts (MEFs), relative to a region of ±2.2 Mb around GEDD boundaries derived from the comparison of Dp1Tyb and wild-type (WT) MEFs. **b** Same elements as in **a** compared relative to TAD boundaries. **c** Density plot showing the distribution of boundaries of TADs from the cortex, boundaries of LADs and RDs from neural precursor cells and CTCF-binding sites from hippocampus relative to GEDD boundaries derived from the comparison of Dp1Tyb and WT hippocampus. **d** Same elements as in **c** compared relative to TAD boundaries

In summary, our results show that GEDDs are not a specific feature of DS, but are detected whenever gene expression changes, and are most likely a direct consequence of clustering of co-expressed genes in the mammalian genome.

## Methods

**Mice**. C57BL/6J.129P2-Dp(16Lipi-Zbtb21)1TybEmcf/Nimr (Dp1Tyb) mice[4] were bred at the MRC National Institute for Medical Research (now part of the Francis Crick Institute). All mice were backcrossed to C57BL/6JNimr for at least ten generations. MEFs were derived from five Dp1Tyb and four WT littermate E14.5 embryos. For the hippocampal RNAseq experiments, the whole hippocampus was dissected from 5 Dp1Tyb and 5 WT littermate male mice aged 18.5–19 weeks. All animal work was approved by the Ethical Review panel of the Francis Crick Institute and was carried out under Project Licences granted by the UK Home Office.

**Cell culture**. For the generation of MEFs, E14.5 embryos were decapitated, eviscerated, minced and then treated with trypsin. Embryos were then titurated to obtain a single-cell suspension. Cells were cultured in Dulbecco's modified Eagle's medium (Gibco), 10% foetal bovine serum, 1× penicillin/streptomycin (Gibco) and 50 μM 2-mercaptoethanol (Gibco). MEFs were cultured for two passages and then prepared for RNA extraction by trypsinisation to remove them from the culture plates.

**RNA preparation, library preparation and sequencing**. Total RNA was purified from MEFs using Trizol (Life Technology) following the manufacturer's instructions. Where RNA spike-ins were used, 4 μl of a 1:100 dilution of ERCC RNA spike-in mix 1 (ThermoFisher), was added to 200,000 MEFs re-suspended in Trizol prior to RNA extraction. The MEF/spike-in/Trizol mixture was then extracted

following the manufacturer's instructions. Hippocampus was homogenised in Qiazol using a TissueRuptor II with disposable probes (Qiagen). RNA was extracted from the homogenised samples using the miRNeasy Kit (Qiagen) followed by treatment with Turbo DNAse (Ambion). RNA concentrations were measured using the Qubit 3 (Life Technologies) or the NanoDrop ND-1000 and RNA quality was assessed using the Bioanalyzer or the Tapestation 2200 (Agilent). Samples with a RNA integrity number >8.5 were taken forward for sequencing. RNAseq libraries were prepared with the TruSeq Stranded mRNA Sample Prep Kit (Illumina). Libraries were sequenced with an Illumina HiSeq 2500 using a 100 base paired-end protocol (MEFs) or with a HiSeq 4000 using a 75 base paired-end protocol (hippocampus) or 100 base paired-end protocol (MEFs with spike-ins).

**Analysis of RNAseq data**. The quality of the sequencing data was assessed using FastQC (http://www.bioinformatics.babraham.ac.uk/projects/fastqc/) and illustrated using the multiqc tool[31] (Supplementary Fig. 15a–c). The adapter sequences were trimmed using TrimGalore! and the reads were mapped to the genome assembly GRCm38 using TopHat (version 2.0.12)[32]. Multialigning reads were discarded. Reads mapping to genes were counted using htseq-count[33]. A R/bioconductor package DESeq2 was used for analysis of differentially expressed genes[34]. Genes were considered as significantly differentially expressed between mutant and WT conditions when $p_{adj} < 0.05$. Data with the ERCC spike-ins was processed as described above with the following differences: data were mapped to the genome assembly amended with sequences of the ERCC synthetic spike-in RNAs (ThermoFisher) and was normalised using spike-ins before analysing differentially expressed genes using a R/Bioconductor package DESeq2. For each RNAseq sample, the total number of reads, as well as the number and percentage of unique and aligned pairs of sequences, are shown in Supplementary Fig. 15d.

For no-genotype-difference analysis, four samples of each genotype were used and their genotypes were mixed so that in each case we compared gene expression in two WT and two mutant samples against the gene expression of two other WT

and two other mutant samples. With 4 WT and 4 mutant samples, there are 18 possible ways to make such comparisons where there is no difference in genotype between the groups of samples being compared; we calculated the expression changes in all such 18 combinations. Expressed genes were filtered using a cutoff value, which was determined by calculating the mean RPKM value for each gene, plotting the $\log_2$ transformed values as a density plot and then fitting a normal distribution to estimate the mean and SD, similar to a previously published approach[35]. Genes with a $\log_2[\text{RPKM}]$ value bigger than mean $- (3 \times \text{SD})$ were considered as expressed. R/Bioconductor was used to calculate all statistical tests (Loess smoothing function and Pearson correlation) and to visualise the data (fold change, correlation and density plots).

**Statistical analysis of GEDDs.** Statistical significance of the existence of GEDDs in chromosomes was determined by using flip number and energy metrics for false discovery rate (FDR) calculations. A flip metric for each chromosome was calculated by applying the following function over all genes in a chromosome:

$$F = \sum_i^N \begin{cases} 1, \text{sign}(h_i) \neq \text{sign}(h_{i-1}) \\ 0, \text{otherwise} \end{cases}$$, where $F$ is total number of flips, $N$ is number of

genes, and $h$ is gene expression fold-change between two conditions taken from RNAseq experiments. A significantly low number of flips in the chromosome indicates the likely presence of GEDDs. The energy metric is defined as a sum of every gene interacting with its neighbouring genes: $E = \sum_i^N h_i \times h_{i-1} + h_i \times h_{i+1}$ where $E$ is total energy. A significantly high energy indicates the likely presence of GEDDs on a chromosome. An FDR calculation was performed by permuting the order of all genes in a given chromosome 100,000 times and then calculating flip and energy metrics for each permutation. This action creates normal distributions of these two metrics for each chromosome. Flip and energy metric values of the original gene order were placed within these distributions and were considered to be significant if they lay >2 SDs from the mean (FDR < 4.55% of event occurring randomly). Hsa21 was excluded from this analysis of human DS fibroblast data and genes in the duplicated region of Mmu16 were excluded from analysis of Dp1Tyb MEFs and hippocampus.

**Alignment of GEDDs with other genomic elements.** A GEDD was defined as a set of two or more expressed, adjacent and similarly regulated (either positive or negative log2[fold change] values) genes. The boundaries of GEDDs were defined as being half way between two genes with opposite directions of fold change since we have no way of defining the location of these boundaries more precisely. The boundaries of GEDDs from MEFs were compared to the boundaries of TADs (Gene Expression Omnibus (GEO): GSE104367)[36], LADs (GEO: GSE17051)[37] and replication domains (GEO: GSM450292)[38], all from MEFs and to CTCF-binding sites from MEFs (GEO: GSE104427)[36]. The boundaries of GEDDs from mouse hippocampus were compared to the boundaries of TADs from the mouse cortex (GEO: GSE35156)[39], to LADs (GEO: GSE17051)[37] and replication domains (GEO: GSM450284)[40] from mouse neural precursor cells and to CTCF-binding sites from the mouse hippocampus (GEO: GSE84174)[41]. For each of these chromatin features, their relative distance to a GEDD or TAD boundary was calculated within a window of ±2.5 Mb around the boundary. The boundary orientation was considered for these two domains. A chromatin feature identified downstream of the start of the GEDD or TAD domain or upstream of the end of the domain was given a positive distance, whereas a feature upstream of the start of the domain or downstream of the end of the domain was given a negative distance. The Gaussian kernel density estimates were computed for the relative distances using the density function with default parameters in the stats R package.

**Numbers of genes.** Numbers of coding genes were determined using Biomart in Ensembl (mouse genome assembly GRCm38.p5) filtering for protein-coding as gene type.

**Reporting summary.** Further information on research design is available in the Nature Research Reporting Summary linked to this article.

## Data availability

All relevant data supporting the key findings of this study are available within the article and its Supplementary Information files or from the corresponding author upon reasonable request. All RNAseq data has been deposited in the Gene Expression Omnibus, accession number GSE109295. A reporting summary for this article is available as a Supplementary Information file.

## Code availability

Scripts for calculating flips and energy are freely available on GitHub (https://github.com/evahelena/GEDDs_paper).

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

## Acknowledgements

We thank Nick Luscombe for helpful discussions, and the Advanced Sequencing Facility and the Biological Research Facility of the Francis Crick Institute for sequencing and animal husbandry, respectively. We thank Rasim Barutcu for TAD boundary data. V.L.J.T. and E.M.C.F. were supported by the Wellcome Trust (grants 080174, 098327 and 098328) and V.L.J.T. was supported by the UK Medical Research Council (Programme U117527252) and by the Francis Crick Institute, which receives its core funding from Cancer Research UK (FC001194), the UK Medical Research Council (FC001194) and the Wellcome Trust (FC001194). J.B. was supported by the Francis Crick Institute, which receives its core funding from Cancer Research UK (FC100051), the UK Medical Research Council (FC100051) and the Wellcome Trust (FC100051). E.P. was supported by the BBSRC (FBAFG 509872).

## Author contributions

N.A., N.D., J.T. and F.W. carried out the experiments. H.A. and E.P. analysed the data. E.L.-E. and S.W.-S. generated the mouse strain used in this work. H.A., N.A. and V.L.J.T. wrote the paper. J.B., K.P., E.M.C.F. and V.L.J.T. supervised the study.

## Additional information

**Competing interests:** The authors declare no competing interests.

