## [Peer Review File · Nature Communications]

Reviewers' comments:

Reviewer #1 (Remarks to the Author):

The manuscript by Anyanwu et al. entitled "Gene expression dysregulation domains (GEDDs) are not a specific feature of Down syndrome" is mainly set to challenge the previous study published by Letourneau et al. that concluded that gene dosage imbalance consecutive to trisomy 21 was associated with global (genome-wide) altered gene expression patterns organized in large GEDDs, corresponding to clusters of up- or down- regulated genes.

The Letourneau's study hinged on analyzing samples from monozygotic twins divergent for trisomy 21. In essence, differential expression analysis comparing those samples identified GEDDs. Then, taking smoothed averaged expression values for all genes between 8 unrelated individuals, Letourneau showed the disomic sample did not exhibit GEDDs, whereas the trisomic sample did, comparing to the smoothed baseline of the 8 individuals.

Those down-syndrome associated GEDDs have been already challenged by Do. et al. 2015 (Do LH, Mobley WC, Singhal N. Questioned validity of Gene Expression Dysregulated Domains in Down's Syndrome. *F1000Research*. 2015; 4:269. doi:10.12688/f1000research.6735.1.), who re-analyzed the Letourneau data, raising the important point that if this GEDD phenomenon would be sound and solid, they should be detectable from any human cohorts of trisomic versus diploid cases, over inter-individual genetic variation, which is not the case.

Anyanwu et al also claim here the GEDDs have nothing to do with DS. I agree with both the comments made by Do et al. and the submitted work of Anyanwu that the Letourneau study has a number of shortcomings. However, RNAseq analysis is a complex topic, the output results depend on the programs used, thresholds, etc. The amplitude of the gene expression changes within a given GEDD is small, with significant noise. Another issue in comparing RNAseq data, is the type, quality and complexity of the libraries constructed and the total number of reads, which is impossible to find out here, making the findings very difficult to assess without re-analysis.

The work by Anyanwu et al. relies on RNAseq data generated on several tissues or cells (MEFs) isolated from a new mouse model developed in their lab, trisomic for 23 Mb of mouse chr. 16 syntenic to human chr. 21. The authors took care of designing rigorous statistical tests looking also into the fold changes in the GEDDs. They also performed analysis by switching genotypes, so that they compared sample pools with identical genotypes, and still observed GEDDs, thus demonstrating that they are not linked to the trisomic phenotype.

Further, they used non-DS related samples (ZFP36L1-deficient follicular B cells versus wild-type cells), and still observed GEDDs. The authors thus concluded that GEDDs are not a specific feature of DS, but a pattern observed whenever any gene expression perturbation occurs. They also discussed that the location of the GEDDs are different for man and mouse experiments. They finally attributed the observed GEDDs to small variations from one culture to the other.

The work presented by Anyanwu et al. mention important issues, which are the analysis and interpretation of gene differential expression data as well as global transcriptome perturbation consecutive to aneuploidy. However, the manuscript seems to hesitate between a technical paper and a review of Letourneau's work, which gives the overall impression of being somewhat unfinished. We finally do not learn much more about the GEDDs, whether they merely reflect stochastic expression, or whether there is an orchestrated response of the genome to any perturbation. The "behavior" of individual genes inside GEDDs across different experiments is not clearly described either.

Further, additional experiments/computing analysis are mandatory to leverage the impact of the work. It would be interesting to compare a number of different genomic perturbations (there should be enough datasets available), to map whether the GEDDs are always located at the same place. Zooming within the GEDDs to estimate the transcriptional response of individual genes, and types of genes, would be interesting. One could also consider to map those domains with TADs to try to rationalize the findings into the global genome architecture.

The manuscript will gain in scope by considering presenting the study in a more global view.

Reviewer #2 (Remarks to the Author):

Recently, in a study by Letourneau et al, it was found that in a human and mouse model (Ts65Dn) of trisomy 21, up- or down-regulated genes were organized in clusters, which were termed Gene Expression Dysregulation Domains (GEDDs).

The present manuscript from Manyawu et al., aimed to replicate those findings in an improved mouse model of trisomy 21 (Dp1Tyb). In a first step, the authors employed the same method to detect GEDDs as published by Letourneau et al. (a Loess smoothed curve) on two RNA-seq datasets generated from their mouse model: MEFs and hippocampus tissue. They also devised statistical tests, to determine significance. Indeed they detected the presence of significant clusters of up- or down-regulated genes, albeit the magnitude of those changes was very small. In a next step they aimed to test if the changes seen in their dataset correlated with the changes reported in the human fibroblasts reported in Letourneau et al., and which were reported to be conserved in the Ts65Dn model. Surprisingly the authors saw only very weak correlation between their de-regulated genes and the previously published data. This inspired the authors to the authors to performed some rigorous tests to assess the validity of their previous findings - the key experiment of the manuscript: they grouped their samples not according to genotype, but randomly grouped trisomy 21 and control samples before running their analysis again. Strikingly, they still detected clusters of up- and down-regulated genes in most of their randomly per-mutated samples. They conclude that the presence of GEDDs is not dependent on trisomy 21. This is further corroborated by their analysis of samples from WT vs. ZFP36L1 mice, where de-regulated genes also seemed to be organized in clusters.

The main content of this manuscript is the reassessment of the findings reported in Letourneau et al. which have been discussed controversially in the field. Using statistical tests and datasets generated from a genetic mouse model of trisomy 21, as well as re-analysing datasets from the original report, they convincingly show that GEDDs are not caused by trisomy 21. This is a very important contribution to the the field, it is of general interest in genetics, and deserves to be published prominently.

Nevertheless, the scope of this study is limited and after establishing that GEDDs are present (but not due to trisomy 21) the manuscript does could have dug deeper to further characterize those GEDDs. There are several potentially interesting questions, for example: how well are the locations of GEDDs conserved between different sample-sets and tissues? Specifically the location of their borders ("flips") might be worth being closely examined: are they marked by CTCF-binding? Do the clusters correlate with lamin-associated or chromatin domains?

Minor comments:

λ□Analogous to Sup Fig 1 the authors also should provide a figure showing an example of gene expression changes and GEDD formation after analysis with randomly per-mutated samples for MEFs and hippocampus.

λ□Fig 7 a and b: it would improve readability if the authors could provide the "correct" blot from Fig 2 for direct comparison.

Reviewer #3 (Remarks to the Author):

In this manuscript, Anyanwu et al. profiled transcription in embryonic fibroblasts and hippocampus derived from their own recently published Dp1Tyb mouse model of Down's Syndrome. By comparing these datasets to transcriptomes of wildtype littermates, they mapped clusters of up- or down-regulated genes throughout the genome, called gene expression dysregulation domains (GEDDs), to confirm previously published findings in a human individual with Down's syndrome and the Ts65Dn mouse model (reference 3 of the manuscript). They created Flip and Energy statistics to assess the significance of GEDDs, and applied these tests to all of the aforementioned datasets and to a set of euploid B cell transcriptomes from transcription factor knockout and wild-type mice. Based on the Flip and Energy statistics, they find clustered changes in gene expression

in all of these datasets and conclude (prematurely based on my comments below) GEDDs are not specific to Down's Syndrome. Overall, I find the topic of study interesting and think it has the potential to either demonstrate or rule out the possibility that the observed GEDDs in reference 3 of the manuscript are technical artifacts. In its current state, the manuscript hasn't convincingly done either but perhaps could with heavy revisions and additional analysis, including comparisons of what the authors consider "GEDDs" in euploid cells to chromosome domains mapped by lamina association, replication timing, and other chromatin features. Below are specific points that the authors should address:

1) The authors state on page 3: "The published study was limited in scope because GEDDs were identified using 4 technical replicates, but only 1 biological replicate of RNA from human DS and euploid fibroblasts, and the analysis of mouse Ts65Dn fibroblasts was carried out using just 1 replicate." The published study used three biological replicates of primary fibroblasts from the same individual that were cultured separately to passage 10 at different times and with RNA extracted from each replicate. The authors should correct the statement and similar statements made throughout the manuscript accordingly. Perhaps the authors intended to point out that a major limitation of the study was the fact that biological replicates from only one human individual were analyzed. I agree this was a major limitation, although this limitation was partially addressed by similar findings from the Ts65Dn mouse model. The next helpful step would be analyzing another rare pair of human monozygotic twins discordant for down syndrome, but that would be more than a biological replicate.

2) The authors state on page 5: "Surprisingly, however, the study did not attempt to determine if the clustering of gene expression changes seen in human DS cells was statistically significant." "Surprisingly" is subjective and should be deleted. Similarly, "attempt to" should be deleted since I assume the authors are not entirely aware of what was attempted by the other group.

3) Referring to their own mapped GEDDs, on page 5 the authors state that "across the genome, the fold changes were small." The authors should show a quantified, side by side comparison of their GEDDs with the previously published ones.

4) Increased expression by 1.5 fold for trisomic regions makes sense, but the significance of genes that are downregulated by 1.5 fold in Down's syndrome or other circumstances where copy number increases from 2 to 3 is unclear. Dosage does not predict this, so it is not obvious why the authors indicate the -1.5 fold threshold in Figures 1 and S1 and S6. The authors should explain this or remove the negative threshold line. Similarly, they should state the number of downregulated significant differentially expressed genes that were located in the duplicated region (it appears to be zero from Figure S1) and clarify whether the average fold changes reported in Figure 1 were calculated from positive/negative fold changes or their absolute values.

5) The flip statistic and accompanying analysis is clear. Flip locations are essentially GEDD boundaries, so flip locations should align with previously mapped chromosome boundaries (e.g. LAD, replication timing, TAD, ridge domains, etc.) if they are the result of domain-wide transcription changes as suggested by the authors and the original GEDD paper. The authors should compare GEDD flip locations to previously mapped chromosome domain boundaries and adjust their conclusions accordingly.

6) Why isn't HSA21 plotted in Figure 6c-d? Why was it excluded from the analysis as stated in the Methods? There are 4 times as many flips in 6b as in 6a, suggesting this chromosome's GEDDs are specific to Down's Syndrome.

7) The energy function is defined on page 6 as "a sum over all genes on the chromosome of products of the fold change of each gene with each of its two neighbours." This language is very unclear and should be reconsidered. The function itself is intelligible by looking at Figure 3a, but the authors don't explain why it is appropriate to multiply fold changes of adjacent genes together rather than taking the average or sum over the sliding 3-gene window, which seems more appropriate to me. The absolute value can be used if the only reason for multiplying is to ensure that clustering of both up- and down-regulated genes affect the statistic in the same direction. The utility of the energy function seems to be the same as the loess smoothed fold change curve.

Multiplying fold changes together is not intuitive and could potentially introduce artifacts, so the energy function should be omitted or modified with a justification and list of potential artifacts stated in the text.

8) Since the original GEDD publication found that GEDDs were masked by natural genetic variation among individuals, the authors should address the possibility that genetic differences between Dp1Tyb littermates could contribute to the decreased magnitude of the expression changes they find relative to the original publication (reference 3 in the manuscript). The Dp1Tyb mice used in this study were the result of 10 backcrosses to C57BL/6JNimr mice so individuals are expected to be 99.902% identical to C57BL/6JNimr, while the rest could have variable contributions from the hm-1 mESCs (derived from 129/ola mice) used to make the Dp1Tyb mice. If the remaining 0.098% is negligible, the authors should demonstrate it using their sequencing data to quantify genetic variation among individuals in their analyzed cohorts and compare this variation to that of the Ts65Dn mice and the 8 unrelated Down's Syndrome individuals analyzed in reference 3.

9) I don't see the point of the comparison on page 7: "We also examined the correlation in gene expression changes between the Dp1Tyb mouse hippocampi and human DS fibroblasts." It is not surprising that transcriptomes of different cell types from different species with different chromosomal abnormalities are not well correlated.

10) On page 8, the authors conclude "the location of [GEDDs] is not conserved between human DS and mouse models", but the authors comparison (fold change on a gene-by-gene basis) is influenced more by direction and magnitude than chromosomal location, so the conclusion is inappropriate. Moreover, they compare mouse hippocampus to human fibroblasts. Due to cell type specificity, the authors need to compare the same tissue in human and mouse to draw inference about evolutionary conservation. Furthermore, conservation of location is to be expected based on conservation of chromosome domains as defined by gene expression, replication timing, subnuclear positioning, association with the nuclear lamina, etc. so this conclusion contradicts the conclusion the authors make elsewhere including on page 12 that "GEDDs are most likely a direct consequence of clusters of co-expressed genes in the mammalian genome" (this later conclusion must be based on assumption and speculation since the GEDDs in this manuscript are not compared to previously mapped chromosome domains).

11) Part of the contradiction between some the authors' conclusions would be alleviated by separating the concept of GEDDs from the established concept of coordinated gene regulation across chromosome domains. The regulation of domains of chromosome structure and function between different cell types and other conditions has been substantiated in many other studies over a number of years. The Flip statistic is more a test of this latter established concept than specifically of GEDDs. The specific aspects of GEDDs that should be tested are a) that GEDDs are found between samples from the same cell type whose only difference is trisomy, and b) that GEDDs are entirely explained by a decreased dynamic range of gene expression in the trisomic condition.

12) The conclusion the authors make on page 8 that the location of GEDDs are not conserved "between different tissues in the Dp1Tyb mouse strain" is expected. There were differences between the fibroblast and iPSC GEDDs from reference 3 and many chromosome domain properties have been demonstrated to be tissue specific. The authors should compare the tissue specific changes they find with tissue specific regulation of chromosome domains as defined by gene expression, replication timing, subnuclear positioning, association with the nuclear lamina, etc.

13) The authors should review and reference a wider scope of literature, especially with regard to chromosome domains to broaden the accessibility of the work to other fields as desired for Nature Journals.

Ben Pope

Reviewer #4 (Remarks to the Author):

This is a very good paper on an important and timely topic. It is well-written and well conducted. I have essentially no criticisms and I think it should be published as-is.

The authors analyse the model of gene expression dysregulation domains (GEDDs) in Down's Syndrome, as recently proposed by LeTourneau et al. They find important omissions in the Letourneau paper, most notably, the absence of some statistical tests. They analyse the previous expression data as well as newly-generated rigorously and show that the apparent expression dysregulation is simply a combination of the expected 1.5 fold increase in the genes with an extra copy, and some other stochastic expression variation. They develop useful statistical tests for the analysis of the clustering of gene expression changes/variation. They find that GEDDs are observed even in comparisons of expression between cells with the same genotype. The GEDD pattern is thus an emergent property of the underlying gene expression clustering in mammalian genomes.

My only, extremely minor, complaint is that "FC" as an acronym for "fold-change" is awkward, and the full words should be used instead.

Response to Reviewers

We thank the reviewers for their helpful and constructive comments. We provide a detailed response to each point below, indicating where we have changed the manuscript and added more data or more analysis.

In addition to these changes, made in response to the reviewers, we also added one further experiment in light of a paper published in the last couple of weeks as we were preparing this revised manuscript. The publication by Mowery *et al* (Cell Reports 2018) reports that pro-B cells grown in IL7 from the bone marrow of Ts1Rhr mice have increased levels of RNA compared to WT control cells. Ts1Rhr mice have a duplication of 33 genes that is entirely contained within the 148 genes duplicated in Dp1Tyb mice. In particular, Mowery *et al* show that the increased RNA level is due to an additional copy of the *Hmgn1* gene, which codes for HMGN1, a nucleosome-binding protein that modulates chromatin compaction. The authors show that this increase in RNA is greater for lower expressed genes compared to higher expressed genes and argue that the GEDDs reported by Letourneau *et al* may be an artefact of using a relative normalisation procedure during RNAseq analysis. Such a normalisation gives the appearance of increased expression of lower expressed genes and decreased expression of higher expressed genes, and, since there is some clustering of genes by expression level, this gives the appearance of GEDDs.

The *Hmgn1* gene is duplicated in Dp1Tyb mice and is overexpressed in Dp1Tyb MEFs, suggesting that the GEDDs we detected may also be a consequence of a global increase in RNA levels. To test this idea, we carried out an RNAseq experiment with spike-in control RNAs and normalised gene expression to these, similar to the approach used by Mowery *et al*. We find a ~1.8% increase in RNA levels in Dp1Tyb MEFs. This is much smaller than the ~10% increase reported by Mowery *et al* in Ts1Rhr pro-B cells. Importantly we can see no preferential increase in expression of lower expressed genes, so this cannot explain the GEDDs we detect in these cells. This data is shown in a new Supplemental Fig. 11, presented on pp9/10 and discussed on p14.

Reviewer #1 (Remarks to the Author):

The manuscript by Anyanwu *et al*. entitled “Gene expression dysregulation domains (GEDDs) are not a specific feature of Down syndrome” is mainly set to challenge the previous study published by Letourneau *et al*. that concluded that gene dosage imbalance consecutive to trisomy 21 was associated with global (genome-wide) altered gene expression patterns organized in large GEDDs, corresponding to clusters of up- or down- regulated genes.

The Letourneau’s study hinged on analyzing samples from monozygotic twins divergent for trisomy 21. In essence, differential expression analysis comparing those samples identified GEDDs. Then, taking smoothed averaged expression values for all genes between 8 unrelated individuals, Letourneau showed the disomic sample did not exhibit GEDDs, whereas the trisomic sample did, comparing to the smoothed baseline of the 8 individuals.

Those down-syndrome associated GEDDs have been already challenged by Do. *et al*. 2015 (Do LH, Mobley WC, Singhal N. Questioned validity of Gene Expression Dysregulated Domains in Down’s Syndrome. F1000Research. 2015;4:269. doi:10.12688/f1000research.6735.1.), who re-analyzed the Letourneau data, raising the important point that if this GEDD phenomenon would be sound and

solid, they should be detectable from any human cohorts of trisomic versus diploid cases, over inter-individual genetic variation, which is not the case.

Anyanwu et al also claim here the GEDDs have nothing to do with DS. I agree with both the comments made by Do et al. and the submitted work of Anyanwu that the Letourneau study has a number of shortcomings. However, RNAseq analysis is a complex topic, the output results depend on the programs used, thresholds, etc. The amplitude of the gene expression changes within a given GEDD is small, with significant noise. Another issue in comparing RNAseq data, is the type, quality and complexity of the libraries constructed and the total number of reads, which is impossible to find out here, making the findings very difficult to assess without re-analysis.

As requested, we provide RNAseq quality details in a new Supplemental Fig. 15. This includes FastQC plots to show the quality of the actual sequencing, as well as data on the numbers of reads, mapped reads, and aligned pairs, as well as numbers and percentages of unique, concordant and aligned pairs. The Methods section already had details on how RNA was purified, libraries made, sequencing machine used, read length, and analysis pipeline.

The reviewers can also directly access the GEO submission of all the RNAseq data. It is currently private, but using the token below, the reviewers can examine the data:

To review GEO accession GSE109295:

Go to <https://www.ncbi.nlm.nih.gov/geo/query/acc.cgi?acc=GSE109295>

Enter token wpgvkqwwfdglryt into the box.

The work by Anyanwu et al. relies on RNAseq data generated on several tissues or cells (MEFs) isolated from a new mouse model developed in their lab, trisomic for 23 Mb of mouse chr. 16 syntenic to human chr. 21. The authors took care of designing rigorous statistical tests looking also into the fold changes in the GEDDs. They also performed analysis by switching genotypes, so that they compared sample pools with identical genotypes, and still observed GEDDs, thus demonstrating that they are not linked to the trisomic phenotype.

Further, they used non-DS related samples (ZFP36L1-deficient follicular B cells versus wild-type cells), and still observed GEDDs. The authors thus concluded that GEDDs are not a specific feature of DS, but a pattern observed whenever any gene expression perturbation occurs. They also discussed that the location of the GEDDs are different for man and mouse experiments. They finally attributed the observed GEDDs to small variations from one culture to the other.

The work presented by Anyanwu et al. mention important issues, which are the analysis and interpretation of gene differential expression data as well as global transcriptome perturbation consecutive to aneuploidy. However, the manuscript seems to hesitate between a technical paper and a review of Letourneau's work, which gives the overall impression of being somewhat unfinished. We finally do not learn much more about the GEDDs, whether they merely reflect stochastic expression, or whether there is an orchestrated response of the genome to any perturbation. The "behavior" of individual genes inside GEDDs across different experiments is not clearly described either.

The changes in gene expression that we detect in Dp1Tyb MEFs and hippocampus are small. To give a better idea of the behaviour of individual genes, we created an

overlay of the gene expression changes across the whole mouse genome for both the Dp1Tyb MEFs and hippocampus, showing both the individual genes and the Loess smoothing curves. This is shown in a new Supplemental Fig. 10. This shows that there is little correlation in the behaviour of individual genes in these two cell types.

Further, additional experiments/computing analysis are mandatory to leverage the impact of the work. It would be interesting to compare a number of different genomic perturbations (there should be enough datasets available), to map whether the GEDDs are always located at the same place. Zooming within the GEDDs to estimate the transcriptional response of individual genes, and types of genes, would be interesting.

Our results show that the GEDDs seen in Dp1Tyb MEFs and hippocampus do not map to the same location. This is shown by the very low correlation of gene expression changes in Fig 4a and the new Supplemental Fig. 10. Furthermore, we saw very little correlation of the GEDDs in these mouse tissues with those reported by Letourneau et al in the human DS fibroblasts – again this is shown in Fig 4a. We note that Letourneau et al argued that the locations of GEDDs were conserved between cell types (DS fibroblasts and iPSCs) and between species (human DS and Ts65Dn mouse fibroblasts). Our data does not support conservation of GEDDs location either between cell types or across species.

One could also consider to map those domains with TADs to try to rationalize the findings into the global genome architecture.

As suggested by this reviewer and the other reviewers, we have investigated the relationship between the GEDDs we detect and a number of genomic features, including TADs, LADs, replication domains and CTCF binding sites. We find that the immediate location around the boundaries of GEDDs has a higher probability to contain boundaries of TADs and CTCF binding sites. The boundaries of replication domains and lamin-associated domains (LADs) were not enriched at the boundaries of GEDDs. Thus GEDDs at least partially map to TADs. This is shown in a new Fig 8, presented on pp12/13 and discussed on p16. Since TADs are known to contain co-regulated genes, we conclude that the GEDDs we detect are a consequence of the TAD organisation of the mammalian genome.

The manuscript will gain in scope by considering presenting the study in a more global view.

We have revised the manuscript considerably, and with the addition of the comparison of GEDDs to TADS, LADs, replication domains and CTCF binding sites we believe that the significance of our findings for Down Syndrome and more broadly, is now much clearer.

Reviewer #2 (Remarks to the Author):

Recently, in a study by Letourneau et al, it was found that in a human and mouse model (Ts65Dn) of trisomy 21, up- or down-regulated genes were organized in clusters, which were termed Gene Expression Dysregulation Domains (GEDDs).

The present manuscript from Manyawu et al., aimed to replicate those findings in an improved mouse model of trisomy 21 (Dp1Tyb). In a first step, the authors employed the same method to detect GEDDs as published by Letourneau et al. (a Loess smoothed curve) on two RNA-seq datasets generated from their

mouse model: MEFs and hippocampus tissue. They also devised statistical tests, to determine significance. Indeed they detected the presence of significant clusters of up- or down-regulated genes, albeit the magnitude of those changes was very small. In a next step they aimed to test if the changes seen in their dataset correlated with the changes reported in the human fibroblasts reported in Letourneau et al., and which were reported to be conserved in the Ts65Dn model. Surprisingly the authors saw only very weak correlation between their de-regulated genes and the previously published data. This inspired the authors to the authors to performed some rigorous tests to assess the validity of their previous findings - the key experiment of the manuscript: they grouped their samples not according to genotype, but randomly grouped trisomy 21 and control samples before running their analysis again. Strikingly, they still detected clusters of up- and down-regulated genes in most of their randomly per-mutated samples. They conclude that the presence of GEDDs is not dependent on trisomy 21. This is further corroborated by their analysis of samples from WT vs. ZFP36L1 mice, where de-regulated genes also seemed to be organized in clusters.

The main content of this manuscript is the reassessment of the findings reported in Letourneau et al. which have been discussed controversially in the field. Using statistical tests and datasets generated from a genetic mouse model of trisomy 21, as well as re-analysing datasets from the original report, they convincingly show that GEDDs are not caused by trisomy 21. This is a very important contribution to the the field, it is of general interest in genetics, and deserves to be published prominently.

Nevertheless, the scope of this study is limited and after establishing that GEDDs are present (but not due to trisomy 21) the manuscript does could have dug deeper to further characterize those GEDDs. There are several potentially interesting questions, for example: how well are the locations of GEDDs conserved between different sample-sets and tissues?

The location of GEDDs is not well conserved between Dp1Tyb MEFs and hippocampus. This is shown by the very low correlation of gene expression changes in Fig 4a and the new Supplemental Fig. 10. Furthermore, we saw very little correlation of the GEDDs in these mouse tissues with those reported by Letourneau et al in the human DS fibroblasts – again this is shown in Fig 4a. We note that Letourneau et al argued that the locations of GEDDs were conserved between cell types (DS fibroblasts and iPSCs) and between species (human DS and Ts65Dn mouse fibroblasts). Our data does not support conservation of GEDDs location either between cell types or across species.

Specifically the location of their borders (“flips”) might be worth being closely examined: are they marked by CTCF-binding? Do the clusters correlate with lamin-associated or chromatin domains?

As suggested by this reviewer and the other reviewers, we have investigated the relationship between the GEDDs we detect and a number of genomic features, including TADs, LADs, replication domains and CTCF binding sites. We find that the immediate location around the boundaries of GEDDs has a higher probability to contain boundaries of TADs and CTCF binding sites. The boundaries of replication domains and lamin-associated domains (LADs) were not enriched at the boundaries of GEDDs. Thus GEDDs at least partially map to TADs. This is shown in a new Fig 8, presented on pp12/13 and discussed on p16. Since TADs are known to contain co-

regulated genes, we conclude that the GEDDs we detect are a consequence of the TAD organisation of the mammalian genome.

Minor comments:

Analogous to Sup Fig 1 the authors also should provide a figure showing an example of gene expression changes and GEDD formation after analysis with randomly per-mutated samples for MEFs and hippocampus.

As requested, we have added two new figures (Supplementary Fig 12 and 13) showing gene expression changes in randomly-permuted 'no genotype difference' analyses of MEFs and hippocampus.

Fig 7 a and b: it would improve readability if the authors could provide the "correct" blot from Fig 2 for direct comparison.

As requested, we have added a 'correct' plot of gene expression changes on Mmu16 in the comparison of Dp1Tyb v WT MEFs (Fig 7a) and hippocampus (Fig 7e). 'Correct' meaning a comparison of Dp1Tyb v WT samples, as opposed to the 'no genotype difference' analyses which are shown in Fig 7b and 7f.

Reviewer #3 (Remarks to the Author):

In this manuscript, Anyanwu et al. profiled transcription in embryonic fibroblasts and hippocampus derived from their own recently published Dp1Tyb mouse model of Down's Syndrome. By comparing these datasets to transcriptomes of wildtype littermates, they mapped clusters of up- or down-regulated genes throughout the genome, called gene expression dysregulation domains (GEDDs), to confirm previously published findings in a human individual with Down's syndrome and the Ts65Dn mouse model (reference 3 of the manuscript). They created Flip and Energy statistics to assess the significance of GEDDs, and applied these tests to all of the aforementioned datasets and to a set of euploid B cell transcriptomes from transcription factor knockout and wild-type mice. Based on the Flip and Energy statistics, they find clustered changes in gene expression in all of these datasets and conclude (prematurely based on my comments below) GEDDs are not specific to Down's Syndrome. Overall, I find the topic of study interesting and think it has the potential to either demonstrate or rule out the possibility that the observed GEDDs in reference 3 of the manuscript are technical artifacts. In its current state, the manuscript hasn't convincingly done either but perhaps could with heavy revisions and additional analysis, including comparisons of what the authors consider "GEDDs" in euploid cells to chromosome domains mapped by lamina association, replication timing, and other chromatin features. Below are specific points that the authors should address:

1) The authors state on page 3: "The published study was limited in scope because GEDDs were identified using 4 technical replicates, but only 1 biological replicate of RNA from human DS and euploid fibroblasts, and the analysis of mouse Ts65Dn fibroblasts was carried out using just 1 replicate." The published study used three biological replicates of primary fibroblasts from the same individual that were cultured separately to passage 10 at different times and with RNA extracted from each replicate. The authors should correct the statement and similar statements made throughout the manuscript accordingly. Perhaps the authors intended to point out that a major limitation of

the study was the fact that biological replicates from only one human individual were analyzed. I agree this was a major limitation, although this limitation was partially addressed by similar findings from the Ts65Dn mouse model. The next helpful step would be analyzing another rare pair of human monozygotic twins discordant for down syndrome, but that would be more than a biological replicate.

In our studies of mouse gene expression, we usually use the term 'biological replicate' to refer to tissues or cell cultures derived from different mice. In other words, we consider the animal to be the biological replicate. In this context we had described the multiple cell cultures derived from a single individual with DS or a single euploid individual as technical replicates, since they all came from the same individual. Using this reasoning, as the reviewer points out, further biological replicates would require analysing further (very rare) pairs of monozygotic twins discordant for trisomy 21.

However, we agree that it is also possible to consider independent cell cultures derived from the same individual as the biological replicate, as the reviewer suggests. We have amended the text to reflect this. Changes are on pages 3 and 15.

2) The authors state on page 5: “Surprisingly, however, the study did not attempt to determine if the clustering of gene expression changes seen in human DS cells was statistically significant.” “Surprisingly” is subjective and should be deleted. Similarly, “attempt to” should be deleted since I assume the authors are not entirely aware of what was attempted by the other group.

The reviewer is correct. We have amended the text on page 5.

3) Referring to their own mapped GEDDs, on page 5 the authors state that “across the genome, the fold changes were small.” The authors should show a quantified, side by side comparison of their GEDDs with the previously published ones.

We have generated a new figure (Supplemental Fig 2a) showing the fold changes in our Dp1Tyb MEF and hippocampus studies and comparing these to the changes in the published Letourneau et al data on human DS fibroblasts. The figure shows that the changes in gene expression are much smaller in our mouse studies. The data is presented on p5.

4) Increased expression by 1.5 fold for trisomic regions makes sense, but the significance of genes that are downregulated by 1.5 fold in Down's syndrome or other circumstances where copy number increases from 2 to 3 is unclear. Dosage does not predict this, so it is not obvious why the authors indicate the -1.5 fold threshold in Figures 1 and S1 and S6. The authors should explain this or remove the negative threshold line.

Of course, the reviewer is correct that there is no meaning to the line indicating a 1.5-fold decrease in expression. These 'negative' lines have been removed from Figures 1, 2, and 5 and Supplementary Figures 1 and 7.

Similarly, they should state the number of downregulated significant differentially expressed genes that were located in the duplicated region (it appears to be zero from Figure S1) and clarify whether the average fold changes reported in Figure 1 were calculated from positive/negative fold changes or their absolute values.

We apologise that this was not clear. We have edited the MA plots and the Table in Figure 1 to make these points more clearly. The MA plots now show the differentially expressed genes ($p_{adj} < 0.05$) as either red dots if they are located in the duplicated region, or as blue dots if located elsewhere in the genome. As now indicated in the table, all the differentially expressed genes in the duplicated region were upregulated. None were downregulated. The average fold change of these genes is calculated from these positive fold changes. We have also clarified this point on pp 5 and 7.

5) The flip statistic and accompanying analysis is clear. Flip locations are essentially GEDD boundaries, so flip locations should align with previously mapped chromosome boundaries (e.g. LAD, replication timing, TAD, ridge domains, etc.) if they are the result of domain-wide transcription changes as suggested by the authors and the original GEDD paper. The authors should compare GEDD flip locations to previously mapped chromosome domain boundaries and adjust their conclusions accordingly.

As suggested by this reviewer and the other reviewers, we have investigated the relationship between the GEDDs we detect and a number of genomic features, including TADs, LADs, replication domains and CTCF binding sites. We find that the immediate location around the boundaries of GEDDs has a higher probability to contain boundaries of TADs and CTCF binding sites. The boundaries of replication domains and lamin-associated domains (LADs) were not enriched at the boundaries of GEDDs. Thus GEDDs at least partially map to TADs. This is shown in a new Fig 8, presented on pp12/13 and discussed on p16. Since TADs are known to contain co-regulated genes, we conclude that the GEDDs we detect are a consequence of the TAD organisation of the mammalian genome.

6) Why isn't HSA21 plotted in Figure 6c-d? Why was it excluded from the analysis as stated in the Methods? There are 4 times as many flips in 6b as in 6a, suggesting this chromosome's GEDDs are specific to Down's Syndrome.

Hsa21 was excluded from the human flips and energy analysis, as was the duplicated region of Mmu16 (Fig 7) since their increased copy number, resulting in increased gene expression strongly skews the flips and energy functions, showing very significant decreases in flips and increases in energy. The aim of the flips and energy analysis was to evaluate the effect of this increased gene dosage on GEDDs in other, non-duplicated regions of the genome, which is why we left them out.

The expectation for Hsa21 is that of course all or most of the genes should be upregulated in DS, because of the extra dosage, i.e. the whole chromosome might appear to be one huge 'GEDD', but this would be an expected consequence of increased dosage. Nonetheless it is instructive to note that while there is some upregulation of genes on Hsa21 in the Letourneau *et al* data in Fig 6a, it does not look even across the chromosome. By contrast, the upregulation of genes in the duplicated region of Mmu16 in Dp1Tyb is much clearer (see Fig 2, 5, 7a and 7e). We suspect that this is due to much larger variation in gene expression in the human data (see Supplementary Fig 2b and the discussion below).

The apparent 4-fold increase in flips in 6b v 6a is not quite correct. The reviewer is probably referring to the curve and the number of times it crosses the 0 line. This curve is the smoothed Loess curve so it's not the same as the flips analysis which uses no smoothing. The flips are counted whenever consecutive genes have gene expression changes in opposite directions. To show these one would need to plot a line between all the genes, with no smoothing. This would have a very jagged

appearance and would cross the 0 line many times. As seen in Fig 6c, 7c and 7g, the number of flips on chromosomes varies from around 200 to over 1000.

7) The energy function is defined on page 6 as “a sum over all genes on the chromosome of products of the fold change of each gene with each of its two neighbours.” This language is very unclear and should be reconsidered. The function itself is intelligible by looking at Figure 3a, but the authors don’t explain why it is appropriate to multiply fold changes of adjacent genes together rather than taking the average or sum over the sliding 3-gene window, which seems more appropriate to me. The absolute value can be used if the only reason for multiplying is to ensure that clustering of both up- and down-regulated genes affect the statistic in the same direction. The utility of the energy function seems to be the same as the loess smoothed fold change curve. Multiplying fold changes together is not intuitive and could potentially introduce artifacts, so the energy function should be omitted or modified with a justification and list of potential artifacts stated in the text.

We apologise that our language was not clear. We have amended the text describing the energy function on page 6 – hopefully it is now clearer.

The reviewer questions the use of the energy function where we multiply fold changes of neighbouring genes, suggesting it does not seem appropriate.

The energy function is a spatial correlation measurement that is well established and used throughout statistical mechanics, as well as in astronomy, financial analysis etc, as a way to measure order. For our purposes, it is robust indicator of spatial clustering in gene expression changes since it takes into account both direction and magnitude of the change. While the Flips metric also provides a measure of clustering of gene expression changes (and generates data consistent with the Energy Function), it is potentially sensitive to small changes in gene expression that could affect our estimates by more frequently identifying reversals in direction of change. The energy function is therefore a robust indicator of spatial clustering in gene expression changes.

8) Since the original GEDD publication found that GEDDs were masked by natural genetic variation among individuals, the authors should address the possibility that genetic differences between Dp1Tyb littermates could contribute to the decreased magnitude of the expression changes they find relative to the original publication (reference 3 in the manuscript). The Dp1Tyb mice used in this study were the result of 10 backcrosses to C57BL/6JNimr mice so individuals are expected to be 99.902% identical to C57BL/6JNimr, while the rest could have variable contributions from the hm-1 mESCs (derived from 129/ola mice) used to make the Dp1Tyb mice. If the remaining 0.098% is negligible, the authors should demonstrate it using their sequencing data to quantify genetic variation among individuals in their analyzed cohorts and compare this variation to that of the Ts65Dn mice and the 8 unrelated Down’s Syndrome individuals analyzed in reference 3.

As requested we have compared the variation in our mouse gene expression data to the human data from Letourneau et al, and show this in a new Supplementary Fig 2b.

We chose not to compare the mouse data to the gene expression data from the 8 unrelated DS individuals that are referred to in Letourneau et al, since this is microarray data, not RNAseq, and we do not think it would be meaningful to compare gene expression variation determined by two very different methods.

Instead we compared the variation of gene expression in our Dp1Tyb MEF and hippocampus studies to the variation in the human fibroblast data derived from the monozygotic twins discordant for trisomy 21, since these were determined using RNAseq. As can be seen in Supplementary Fig 2b, there is much more variation in the human data compared to our mouse data. This is probably the reason for the much larger fold changes in gene expression in the human data (Supplementary Fig 2a). We note that this human data is from isogenic cultures that all came from the same individual, whereas our data is from multiple animals, albeit backcrossed for >10 generations. This suggests that the large variation in the human data seen within each genotype separately is most likely due to substantial experimental noise – culture conditions, RNA prep, library construction, etc.

9) I don't see the point of the comparison on page 7: "We also examined the correlation in gene expression changes between the Dp1Tyb mouse hippocampi and human DS fibroblasts." It is not surprising that transcriptomes of different cell types from different species with different chromosomal abnormalities are not well correlated.

We agree that we would also expect the transcriptomes of different cell types and species to be substantially different. However, the Letourneau et al paper explicitly concludes that the location of GEDDs was conserved between human DS fibroblasts and DS iPSCs (2 different cell types) and also conserved between human DS fibroblasts and Ts65Dn mouse fibroblasts (2 different species), i.e. they claim GEDDs are conserved across cell types and species. In that light it made sense for us to look at the potential conservation of GEDDs between cell types (mouse fibroblasts v mouse hippocampus), between species (human fibroblasts v mouse fibroblasts) and both (human fibroblasts v mouse hippocampus). Our conclusion from these comparisons in Fig 4a is that there is very little conservation of GEDDs between cell types or species.

10) On page 8, the authors conclude "the location of [GEDDs] is not conserved between human DS and mouse models", but the authors comparison (fold change on a gene-by-gene basis) is influenced more by direction and magnitude than chromosomal location, so the conclusion is inappropriate. Moreover, they compare mouse hippocampus to human fibroblasts. Due to cell type specificity, the authors need to compare the same tissue in human and mouse to draw inference about evolutionary conservation.

If there is conservation of GEDDs by location between human and mouse data, this should be visible using the gene by gene correlation comparison, since human and mouse genomes share extensive regions of synteny with orthologous genes arranged in the same order. This is exactly what Letourneau et al did in their Fig 4b, top panel in which they compare data from human and mouse fibroblasts, and on the basis of a correlation coefficient of $\rho = 0.44$ conclude that there is conservation of the location of GEDDs between species.

The reviewer asks that we compare the same tissue between human and mouse models. We have done exactly that in Fig 4a (left panel) where we compare gene expression changes between human and mouse fibroblasts and find almost no correlation ($\rho = 0.087$).

Furthermore, conservation of location is to be expected based on conservation of chromosome domains as defined by gene expression, replication timing, subnuclear positioning, association with the nuclear lamina, etc. so this conclusion contradicts the conclusion the authors make elsewhere including on page 12 that "GEDDs are most likely a direct consequence of clusters of co-

expressed genes in the mammalian genome” (this later conclusion must be based on assumption and speculation since the GEDDs in this manuscript are not compared to previously mapped chromosome domains).

As discussed above, in response to suggestions from the reviewers, we have compared the location of GEDDs to other genomic elements and find that the immediate vicinity of GEDDs boundaries is enriched in TAD boundaries and CTCF binding sites, suggesting that GEDDs in part reflect the TAD organisation of the genome. The reviewer is concerned that we do not see conservation of the location of the GEDDs even though they appear to correspond in part to TADs; this might have been expected since the location of TADs is conserved between cell types. We believe this apparent discrepancy is because the gene expression changes we see in the MEFs and the hippocampus are small in number and small in magnitude. Thus, there is not much signal that could give a strong correlation between the cell types (we see a correlation coefficient between MEFs and hippocampus of $\rho = 0.139$, see Fig 4a). If different sets of genes are dysregulated between these two cell types, then this would explain the lack of correlation of gene expression changes. Nonetheless the GEDDs in Dp1Tyb MEFs and hippocampus could still partly align to TADs, with different sets of dysregulated genes aligning to different TADs in the two cell types. In support of this suggestion, we note that excluding genes in the duplicated region of Dp1Tyb, there is not one gene in common between the significantly ($\text{padj} < 0.05$) differentially expressed genes in Dp1Tyb MEFs and hippocampus.

11) Part of the contradiction between some the authors' conclusions would be alleviated by separating the concept of GEDDs from the established concept of coordinated gene regulation across chromosomes domains. The regulation of domains of chromosome structure and function between different cell types and other conditions has been substantiated in many other studies over a number of years. The Flip statistic is more a test of this latter established concept than specifically of GEDDs. The specific aspects of GEDDs that should be tested are a) that GEDDs are found between samples from the same cell type whose only difference is trisomy, and b) that GEDDs are entirely explained by a decreased dynamic range of gene expression in the trisomic condition.

We agree with the reviewer that his points a) and b) are key features of the GEDDs described by Letourneau et al – that GEDDs are found when samples differ by trisomy 21, and that they are explained by a decreased dynamic range of gene expression. We have tested both of these points.

a) We have shown GEDDs in a comparison of Dp1Tyb and WT MEFs and in a comparison of Dp1Tyb and WT hippocampus. However, we can also detect GEDDs when comparing gene expression between samples that do not differ in a DS genotype, e.g. when we use the 'no genotype difference' comparisons, or when we analyse a completely unrelated dataset (WT v ZFP36L1-deficient B cells). Thus, GEDDs are not specific to trisomy 21.

b) We investigated whether GEDDs can be explained by a decreased dynamic range of gene expression. Letourneau et al show that in DS fibroblasts there was an increase in expression of lowly expressed genes and a decrease in the expression of highly expressed genes. We found no evidence for this in Dp1Tyb MEFs or Dp1Tyb hippocampus. This is shown in a new Fig 4b and presented on pp8/9.

Taken together our results show that the clustered gene expression changes we detect in Dp1Tyb MEFs and hippocampus are not specific to DS and are not caused by a decreased dynamic range of gene expression.

12) The conclusion the authors make on page 8 that the location of GEDDs are not conserved “between different tissues in the Dp1Tyb mouse strain” is expected. There were differences between the fibroblast and iPSC GEDDs from reference 3 and many chromosome domain properties have been demonstrated to be tissue specific.

Letourneau et al compared the GEDDs they detected in human DS fibroblasts with those in human DS iPSCs and reported substantial similarity between these two different cell types ($\rho = 0.85$) – see p347 and Fig 3 of Letourneau et al. We note the title of the second paragraph on this page - “GEDDs are conserved in induced pluripotent stem cells”. Thus, based on this publication, we expected to find correlation in GEDDs between cell types. However, we found only very low correlation (Fig 4a in our manuscript).

The authors should compare the tissue specific changes they find with tissue specific regulation of chromosome domains as defined by gene expression, replication timing, subnuclear positioning, association with the nuclear lamina, etc.

As described above, we compared the location of GEDDs to several genomic elements: TADs, LADs, replication domains and CTCF binding sites, and report the results in a new Fig 8.

13) The authors should review and reference a wider scope of literature, especially with regard to chromosome domains to broaden the accessibility of the work to other fields as desired for Nature Journals.

As suggested, we have added references to other genomic domains, specifically to TADs, LADs, replication domains and CTCF binding sites, whose location we now compare to GEDDs.

Reviewer #4 (Remarks to the Author):

This is a very good paper on an important and timely topic. It is well-written and well conducted. I have essentially no criticisms and I think it should be published as-is.

The authors analyse the model of gene expression dysregulation domains (GEDDs) in Down's Syndrome, as recently proposed by LeTourneau et al. They find important omissions in the Letourneau paper, most notably, the absence of some statistical tests. They analyse the previous expression data as well as newly-generated rigorously and show that the apparent expression dysregulation is simply a combination of the expected 1.5 fold increase in the genes with an extra copy, and some other stochastic expression variation. They develop useful statistical tests for the analysis of the clustering of gene expression changes/variation. They find that GEDDs are observed even in comparisons of expression between cells with the same genotype. The GEDD pattern is thus an emergent property of the underlying gene expression clustering in mammalian genomes.

My only, extremely minor, complaint is that "FC" as an acronym for "fold-change" is awkward, and the full words should be used instead.

As requested, we have changed the text throughout to use 'fold change' in full.

REVIEWERS' COMMENTS:

Reviewer #1

Final review of the manuscript NCOMMS-18-06239B

"Gene Expression Dysregulation Domains are not a Specific Feature of Down Syndrome" by H. Ahlfors et al.

In the revised version of the manuscript, the authors have carefully considered and addressed the reviewer's comments, in particular in assessing the location of the GEDD in the more global context of the overall chromosomal architecture such as TADs, LADs, etc.

The analysis of gene expression changes consecutive to aneuploidy is a very interesting topic, despite the inherent challenges encountered in measuring expression changes of limited amplitude in large chromosomal regions. Although the ground truth might be difficult to reach with the current technologies and analysis methods at hand, the authors present a thorough analysis and convincing data in this difficult field. The manuscript sheds light for understanding the consequences of human trisomy 21 on gene expression, and will also provide useful guidance for the analysis of other constitutive or somatic aneuploidies. I therefore recommend publication of the manuscript by H. Ahlfors et al. in Nature Communications.

Reviewer #2 (Remarks to the Author):

The authors have, in the revised version, addressed most of the critical points raised. This reviewer's concerns have been answered successfully.

Reviewer #3 (Remarks to the Author):

The authors have addressed my concerns in the revised manuscript. I don't need to see another version. Below are minor points the authors should consider before publication:

- 1) In the legend to Figure S7 the authors state: "Dashed line on Mmu16 indicates a fold change." The authors need to insert "1.5" before "fold change".
- 2) It would be helpful to add correlation values to each plot in Figure S10.
- 3) It would be helpful to perform a significance test and add the p value to Figure S11.
- 4) On page 10, it would be helpful to more clearly explain the "no genotype difference" analysis and to clearly define the "other" sample in the text and in the legends to Figures S12 and S13. It is unclear since the term "fake genotype" rather than "other" is used in the methods and neither are explicitly defined. Perhaps a schematic would help. It would also be helpful to explain why the fold changes in Figure S13 (hippocampi) so much smaller than in S12 (MEFs).
- 5) On page 6, it would be helpful to explain why the energy function is appropriate for this analysis and add a citation for the method from which the function was adapted.
- 6) On page 8, concluding that GEDDs are not conserved between humans and mouse models seems to contradict the conclusion that GEDDs and TADs are correlated since TADs are largely conserved between human and mouse. It would be helpful to readers if these points were reconciled.
- 7) On page 10, it would be helpful to point out the increase in genetic material (as a percentage) in Dp1Tyb mice to compare this number to the 1.8% increase in total RNA that you observed after absolute normalization. Limiting the comparison to transcribed genetic material would be even better.

8) On page 12, it would be helpful to point out that the distance between genes at the edges of adjacent GEDDs is arbitrary, so using the half-way point is an imprecise way to call GEDD boundaries. Using the actual positions of genes at the edges of GEDDs would be a more precise way to call boundaries and I'd expect this definition to improve alignment to TADs, LADs, RDs, etc. I'd also expect transcriptionally active GEDDs (high transcriptional activity) to have better alignment.

Ben Pope

Response to Reviewers

We thank the reviewers for their helpful and constructive comments. We provide a detailed response to the remaining points made by reviewer 3 below.

Reviewer #3 (Remarks to the Author):

1) In the legend to Figure S7 the authors state: “Dashed line on Mmu16 indicates a fold change.” The authors need to insert “1.5” before “fold change”.

Done

2) It would be helpful to add correlation values to each plot in Figure S10.

Done

3) It would be helpful to perform a significance test and add the p value to Figure S11.

Done

4) On page 10, it would be helpful to more clearly explain the “no genotype difference” analysis and to clearly define the “other” sample in the text and in the legends to Figures S12 and S13. It is unclear since the term “fake genotype” rather than “other” is used in the methods and neither are explicitly defined. Perhaps a schematic would help. It would also be helpful to explain why the fold changes in Figure S13 (hippocampi) so much smaller than in S12 (MEFs).

The main text on p11 and the methods on p19 have been edited to make the nature of this comparison clearer. We have removed the term ‘fake genotype’.

Supplementary Figures 12 (MEFs) and 13 (hippocampus) show an example of the fold changes in gene expression across the genome in one of multiple ‘no genotype comparisons’ that we carried out, in which the changes are calculated by comparing expression in 2 WT and 2 Dp1Tyb samples against 2 other WT and 2 other Dp1Tyb samples. There were 18 such possible combinations that were calculated, and one of these is being shown in each of Supplementary Figures 12 and 13. In these figures the fold changes are not a result of genotype differences between the two sets of data being compared, but instead are derived from other differences in expression between samples, i.e. noise. Since the MEF RNAseq data is noisier than the hippocampal data (Supplementary Figure 2b), the fold changes seen in the MEF comparisons are larger than those in the hippocampal comparisons.

5) On page 6, it would be helpful to explain why the energy function is appropriate for this analysis and add a citation for the method from which the function was adapted.

We have edited the text on p6 to make clear that this is a spatial correlation measure, and thus measures a key aspect that is claimed for GEDDs – clustering of gene expression changes. We have also added a citation.

6) On page 8, concluding that GEDDs are not conserved between humans and mouse models seems to contradict the conclusion that GEDDs and TADs are

correlated since TADs are largely conserved between human and mouse. It would be helpful to readers if these points were reconciled.

The human and mouse fibroblast gene expression changes show very weak correlation, but it is not zero ($r=0.0874$). The edges of mouse GEDDs are enriched at TAD boundaries and vice-versa, but this enrichment does not mean that every GEDD boundary is at TAD boundary. Indeed, it could be that just a small fraction of the GEDDs line up with TAD boundaries, and this then accounts for the poor correlation of fold changes between human and mouse data. Furthermore, the substantially greater noise in the human data compared to the mouse data (Supplementary Figure 2b) could be reducing some of the human-mouse correlation.

7) On page 10, it would be helpful to point out the increase in genetic material (as a percentage) in Dp1Tyb mice to compare this number to the 1.8% increase in total RNA that you observed after absolute normalization. Limiting the comparison to transcribed genetic material would be even better.

Calculations shows that the genome is 0.84% bigger in Dp1Tyb cells than in WT cells, whereas the transcriptome has increased around 0.7% in Dp1Tyb cells compared to WT cells. This increase may account for some of the 1.8% increase in overall RNA level in the Dp1Tyb cells. We discuss this point in the text on p10.

8) On page 12, it would be helpful to point out that the distance between genes at the edges of adjacent GEDDs is arbitrary, so using the half-way point is an imprecise way to call GEDD boundaries. Using the actual positions of genes at the edges of GEDDs would be a more precise way to call boundaries and I'd expect this definition to improve alignment to TADs, LADs, RDs, etc. I'd also expect transcriptionally active GEDDs (high transcriptional activity) to have better alignment.

The analysis of the location of GEDDs was limited to expressed genes, defined as described in the Methods. The location of the boundary of GEDDs is indeed somewhat arbitrary since all we have to go on is the change in direction of the fold change. We have no way of knowing where the boundary of the GEDDs element lies within the gap. In view of this, we chose the half-way point. We have edited the text on p20 to point this out.